# Branch Point Twist Field Form Factors in the sine-Gordon Model I: Breather Fusion and Entanglement Dynamics

Olalla A. Castro-Alvaredo$^\heartsuit$ and Dávid X. Horváth$^\spadesuit$

$^\heartsuit$ Department of Mathematics, City, University of London, 10 Northampton Square EC1V 0HB, UK

$^\spadesuit$ SISSA and INFN Sezione di Trieste, via Bonomea 265, 34136 Trieste, Italy

The quantum sine-Gordon model is the simplest massive interacting integrable quantum field theory whose two-particle scattering matrix is generally non-diagonal. As such, it is a model that has been extensively studied, especially in the context of the bootstrap programme. In this paper we compute the form factors of a special local field known as the branch point twist field, whose correlation functions are building blocks for measures of entanglement. We consider the attractive regime where the theory possesses a particle spectrum consisting of a soliton, an antisoliton (of opposite $U(1)$ charges) and several (neutral) breathers. In the breather sector we exploit the fusion procedure to compute form factors of heavier breathers from those of lighter ones. We apply our results to the study of the entanglement dynamics after a small mass quench and for short times. We show that in the presence of two or more breathers the von Neumann and Rényi entropies display undamped oscillations in time, whose frequencies are proportional to the even breather masses and whose amplitudes are proportional to the breather's one-particle form factor.

**Keywords:** sine-Gordon Model, Integrability, Form Factors, Branch Point Twist Fields, Entanglement Dynamics

$^\heartsuit$ o.castro-alvaredo@city.ac.uk
$^\spadesuit$ david.horvath@sissa.it

31st March 2021

# 1  Introduction

The quantum relativistic sine-Gordon model is a paradigmatic example of an integrable quantum field theory (IQFT) that is amenable to solution by the bootstrap programme. It provides the simplest example of a theory that is interacting and has a non-diagonal $S$-matrix, famously obtained in [1]. This means that the theory allows for backscattering or, in a different language, the $S$-matrix is a non-trivial solution of the Yang-Baxter equation. The theory has a rich particle spectrum containing two fundamental particles known as the soliton ($s$) and the antisoliton ($\bar{s}$) and a tower of breathers ($b_k$) which can be interpreted both as soliton-antisoliton bound states and as bound states of lighter breathers. The number and masses of these breathers depend on the model's coupling constant. Although the theory is non-diagonal in the standard scattering matrix sense, the breather sector is diagonal and this simplifies form factor computations considerably. In addition, in a certain coupling constant regime, the sine-Gordon model can be seen as the continuum limit of another paradigmatic integrable theory, namely the spin-$\frac{1}{2}$ XXZ quantum spin chain.

In the context of the bootstrap programme for IQFTs, the matrix elements of local operators (e.g. form factors) of the sine-Gordon model have been extensively studied by many authors employing many different techniques. Some of the earliest results are due to F.A. Smirnov [2,3], while a long series of papers by the Berlin group employed integral representations and nested Bethe ansatz as solution techniques [4–8]. A different approach known as free field representation was employed in [9,10] and the fermionic structure of the model was exploited in [11,12]. Of particular interest to us is the work [10] which focused on breather form factors and used the fusion technique in order to obtain form factors of heavier breathers from those of the lightest one. There has also been intense study and a large body of applications of sine-Gordon form factors in various other contexts such as the case of finite temperature one-point functions [13,14], quantum quenches [15–17] and boundary field theory [18] and, in particular, in finite volume [19–21] where once again fusion techniques can be employed.

Finally, it is important to note that many studies of the breather form factors (particularly those where fusion is used) exploit the relationship between the sine-Gordon and sinh-Gordon theories. At Lagrangian level the two theories are identical up to the complexification of the coupling constant. In addition, the two-particle $S$-matrix of the first (lightest) breather is mapped to the two-particle scattering matrix of the sinh-Gordon particle under the same transformation. This implies that the form factors of the first breather (and by fusion, also those of higher breathers) can be obtained from those of the sinh-Gordon field by simply changing the coupling constant dependence. Then the sinh-Gordon form factors computed in various papers [22,23] become the starting point of computations in the sine-Gordon model.

The works we have referred to so far are concerned with "standard" local fields of the sine-Gordon theory, such as the sine-Gordon field $\varphi$, its powers and, especially, exponential fields of the form $e^{ia\varphi}$ which are of particular interest as they are related to the trace of the stress-energy tensor. In the present work our main aim is to generalize these results to branch point twist fields, starting with the branch point twist field and associated form factor programme introduced in [24], and then continuing (in part II) with the symmetry resolved branch point twist field recently introduced in [25,26]. Twist field form factors of the sine-Gordon model where first studied in [27] but only in the so-called repulsive regime where no breathers are present. In

this paper we extend those results to the situation when several breathers are present focussing on all non-vanishing one- and two-particle form factors. In the breather sector we employ the results of [24] and [28] where the two- and four-particle form factors of the sinh-Gordon field where obtained, respectively. These will constitute our starting point when employing the fusion procedure to obtain lower particle form factors of higher breathers.

The paper is organized as follows: In Section 2 we review some general results for the sine-Gordon model, notably its $S$-matrix and particle spectrum. In Section 3 we review the definition of the branch point twist field and the main equations satisfied by its form factors. In Section 4 we diagonalize the two-particle form factor equations to compute the two-particle solition-antisoliton form factor. We put special emphasis on the discussion of its dynamical pole structure. In Section 5 we use fusion to compute one- and two-particle breather form factors, including up to four breathers and carry out some simple consistency checks of our solutions. In Section 6 we evaluate the $\Delta$ sum rule in several coupling regimes, finding very good agreement with the exact value of the branch point twist field conformal dimension for all coupling choices. In Section 7 we discuss one application of our results to the study of the entanglement dynamics following a mass quench. We conclude in Section 8. The more technical details of our work are presented in various Appendices. Appendix A summarizes some useful formulae for the minimal form factors. Appendices B and C give details of the computation of breather form factors for the branch point twist field and the trace of the stress-energy tensor, respectively. In both cases we use the fusion procedure. In Appendix D we analyse in more detail the dynamical pole axiom for the soliton-antisolition form factors. In Appendix E we present some additional numerical results concerning our evaluation of the the $\Delta$ sum rule.

## 2 Main Features of the Model

The sine-Gordon model is characterized by the following euclidean action

$$\mathcal{A} = \int dx dt \left[ \frac{1}{16\pi} \left[ (\partial_0 \varphi)^2 - (\partial_1 \varphi)^2 \right] - 2\mu \cos(g\varphi) \right], \tag{1}$$

where $g$ and $\mu$ are coupling constants and $\varphi$ is a scalar field. As anticipated in the introduction, this action becomes that of another theory, know as sinh-Gordon model under the mapping $g \mapsto ig$ with $g \in \mathbb{R}$. For generic values of the coupling, the theory has a rich particle spectrum consisting of a soliton ($s$) and anti-solition ($\bar{s}$) of opposite $U(1)$ charge and a family of bound states known as breathers. Defining the new coupling

$$\xi = \frac{g^2}{1 - g^2}, \tag{2}$$

we have that the masses of the breathers take the form

$$m_k = 2m \sin \frac{\pi k \xi}{2} \quad \text{for} \quad k = 1, 2, \dots, \ell(\xi), \tag{3}$$

where $m$ is the mass of the soliton and the anti-soliton and $\ell(\xi) = \frac{1}{\xi} - 1$ if $\frac{1}{\xi} \in \mathbb{Z}$ and $[\frac{1}{\xi}]$ otherwise, where $[\cdot]$ denotes the integer part. The mass $m$ is related to the couplings $\mu$ and $g$

through the mass-coupling relation

$$\mu = \frac{\Gamma(g^2)}{\pi\Gamma(1-g^2)}\left[\frac{m\sqrt{\pi}\Gamma(\frac{1}{2-2g^2})}{2\Gamma(\frac{g^2}{2-2g^2})}\right]^{2-2g^2}, \tag{4}$$

first found in [29]. There are various interesting regimes:

- For $\xi > 1$ there are no bound states and the full spectrum consist only of the solition and the antisoliton. This is called the *repulsive regime*. In this regime, the theory is equivalent to the *massive Thirring model*, a perturbation of the massive Dirac theory that preserves the $U(1)$ symmetry. We studied the entanglement entropy in this particular regime in [27].

- The point $\xi = 1$ is special as can be seen more precisely from the $S$-matrices given below. From (5) we have that $S_{ss}^{ss}(\theta) = S_{\bar{s}\bar{s}}^{\bar{s}\bar{s}}(\theta) = -1$ and also $S_{s\bar{s}}^{s\bar{s}}(\theta) = -1$ and $S_{s\bar{s}}^{\bar{s}s}(\theta) = 0$. At this point the theory becomes a *Dirac free fermion*.

- For $\xi < 1$ the model is in the *attractive regime* were bound states (breathers) are formed with the masses (3).

- In particular, whenever $\frac{1}{\xi} = n$, with $n \in \mathbb{Z}^+$ the non-diagonal scattering amplitude $S_{s\bar{s}}^{\bar{s}s}(\theta) = 0$ is vanishing and the theory becomes diagonal. In fact, it reduces to the $D_n$-minimal Toda field theory.

The $S$-matrices are [1]

$$
\begin{aligned}
S_{ss}^{ss}(\theta) = S_{\bar{s}\bar{s}}^{\bar{s}\bar{s}}(\theta) &= -\exp\left[-i\int_0^\infty \frac{dt}{t}\frac{\sinh\frac{\pi t(1-\xi)}{2}\sin(t\theta)}{\sinh\frac{\pi t\xi}{2}\cosh\frac{\pi t}{2}}\right] \\
&= \prod_{k=0}^\infty \frac{\Gamma\left(\frac{2k+1}{\xi}-\frac{i\theta}{\pi\xi}+1\right)\Gamma\left(\frac{2k+1}{\xi}-\frac{i\theta}{\pi\xi}\right)\Gamma\left(\frac{2k}{\xi}+\frac{i\theta}{\pi\xi}+1\right)\Gamma\left(\frac{2k+2}{\xi}+\frac{i\theta}{\pi\xi}\right)}{\Gamma\left(\frac{2k}{\xi}-\frac{i\theta}{\pi\xi}+1\right)\Gamma\left(\frac{2k+2}{\xi}-\frac{i\theta}{\pi\xi}\right)\Gamma\left(\frac{2k+1}{\xi}+\frac{i\theta}{\pi\xi}\right)\Gamma\left(\frac{2k+1}{\xi}+\frac{i\theta}{\pi\xi}+1\right)},
\end{aligned} \tag{5}
$$

and

$$S_{s\bar{s}}^{s\bar{s}}(\theta) = S_{\bar{s}s}^{\bar{s}s}(\theta) = \frac{\sinh\frac{\theta}{\xi}}{\sinh\frac{i\pi-\theta}{\xi}}S_{ss}^{ss}(\theta), \qquad S_{s\bar{s}}^{\bar{s}s}(\theta) = S_{\bar{s}s}^{s\bar{s}}(\theta) = \frac{\sinh\frac{i\pi}{\xi}}{\sinh\frac{i\pi-\theta}{\xi}}S_{ss}^{ss}(\theta) \tag{6}$$

where $S_{\bar{s}s}^{s\bar{s}}(\theta)$ and $S_{s\bar{s}}^{\bar{s}s}(\theta)$ are the off-diagonal amplitudes. Useful linear combinations are

$$S_+(\theta) = S_{\bar{s}s}^{s\bar{s}}(\theta) + S_{\bar{s}s}^{s\bar{s}}(\theta) \qquad S_-(\theta) = S_{\bar{s}s}^{s\bar{s}}(\theta) - S_{\bar{s}s}^{s\bar{s}}(\theta). \tag{7}$$

The remaining $S$-matrices are diagonal and can be expressed in terms of the standard blocks:

$$[x]_\theta = \frac{\tanh\frac{1}{2}(\theta+i\pi x)}{\tanh\frac{1}{2}(\theta-i\pi x)}. \tag{8}$$

For instance

$$S_{sb_1}(\theta) = \left[\frac{1+\xi}{2}\right]_\theta, \quad S_{b_1 b_1}(\theta) = [\xi]_\theta, \quad S_{b_2 b_2}(\theta) = [\xi]_\theta^2 [2\xi]_\theta, \tag{9}$$

$$S_{b_1 b_3}(\theta) = [\xi]_\theta [2\xi]_\theta, \quad S_{b_1 b_2}(\theta) = \left[\frac{\xi}{2}\right]_\theta \left[\frac{3\xi}{2}\right]_\theta. \tag{10}$$

An important property of these $S$-matrices is that they have poles in the physical sheet which can be attributed to the presence of a bound state. The residue of such poles plays a role in later sections and so we report some of these results here. In general, we define

$$-i \operatorname*{Res}_{\theta = i\pi u_{ab}^c} S_{ab}(\theta) := (\Gamma_{ab}^c)^2, \tag{11}$$

where $i\pi u_{ab}^c$ is the pole of the $S$-matrix corresponding to the formation of a bound state $c$ in the scattering process $a + b \mapsto c$. This equation provides a definition of the "pole strength" $\Gamma_{ab}^c$. For the $S$-matrices above we have for instance,

$$
\begin{aligned}
\Gamma_{s\bar{s}}^{b_1} &= \sqrt{2\cot\frac{\pi\xi}{2}} \\
\Gamma_{s\bar{s}}^{b_2} &= \sqrt{\frac{1}{4}\sin 2\pi\xi \csc^2 \frac{\pi\xi}{2}} \\
\Gamma_{s\bar{s}}^{b_3} &= \sqrt{2\cot\frac{3\pi\xi}{2}\cot\frac{\pi\xi}{2}\cot\pi\xi} \\
\Gamma_{s\bar{s}}^{b_4} &= \sqrt{2\cot 2\pi\xi \cot\frac{\pi\xi}{2}\cot\pi\xi\cot\frac{3\pi\xi}{2}}.
\end{aligned}
\tag{12}
$$

which can be obtained using the infinite product representation (5). The above quantities are associated with the pole strengths of $S_{s\bar{s}}^{s\bar{s}}(\theta)$ and $S_{ss}^{ss}(\theta)$ for the first few breathers and the position of the poles are at $i\pi\xi k$ with $k = 1, \ldots, \ell(\xi)$ as defined in (3) and assuming we are in the attractive regime. For the breather $S$-matrices, we have

$$\Gamma_{b_1 b_1}^{b_2} = \sqrt{2\tan\pi\xi}, \quad \Gamma_{b_2 b_2}^{b_4} = \frac{2\cos\pi\xi + 1}{2\cos\pi\xi - 1}\sqrt{2\tan 2\pi\xi}, \quad \Gamma_{b_1 b_2}^{b_3} = \sqrt{\frac{2\cos\pi\xi + 1}{2\cos\pi\xi - 1}}\Gamma_{b_1 b_1}^{b_2}, \tag{13}$$

and $\Gamma_{b_1 b_3}^{b_4} = \Gamma_{b_2 b_2}^{b_4}/\Gamma_{b_1 b_2}^{b_3}$. Note that, as mentioned earlier, $S_{b_1 b_1}(\theta)$ coincides with the sinh-Gordon $S$-matrix under the replacement $B = -2\xi$, where $B$ is the sinh-Gordon coupling constant [30,31]. More generally, the following integral formulae hold

$$S_{sb_k}(\theta) = (-1)^k \exp\left[-i\int_0^\infty \frac{dt}{t} \frac{2\cosh\frac{\pi t\xi}{2}\sinh\frac{\pi tk\xi}{2}\sin(t\theta)}{\sinh\frac{\pi\xi t}{2}\cosh\frac{\pi t}{2}}\right]. \tag{14}$$

$$S_{b_k b_p}(\theta) = \exp\left[-i\int_0^\infty \frac{dt}{t} \frac{4\cosh\frac{\pi t\xi}{2}\sinh\frac{\pi tk\xi}{2}\cosh\frac{\pi t(1-\xi p)}{2}\sin(t\theta)}{\sinh\frac{\pi\xi t}{2}\cosh\frac{\pi t}{2}}\right]. \tag{15}$$

for $k < p$ and, finally

$$S_{b_k b_k}(\theta) = -\exp\left[-i \int_0^\infty \frac{dt}{t} \frac{2\left[\cosh\frac{\pi t \xi}{2} \sinh\frac{\pi t(2k\xi-1)}{2} + \sinh\frac{(1-\xi)\pi t}{2}\right] \sin(t\theta)}{\sinh\frac{\pi \xi t}{2} \cosh\frac{\pi t}{2}}\right]. \tag{16}$$

A good summary of all the $S$-matrices, and of how to derive Gamma-function representations from integral representations can be found for instance in [4].

## 3 Branch Point Twist Fields in a Nutshell

It has been known for some time that several entanglement measures, including the Rényi entropies, can be expressed in terms of correlation functions of a special class of local fields $\mathcal{T}$ which have been termed branch point twist fields in [24]. Branch point twist fields are, on the one hand, twist fields in the broader sense, that is, fields associated with an internal symmetry of the theory under consideration [24], and on the other hand related to branch points of multi-sheeted Riemann surfaces [32]. They are twist fields associated to the cyclic permutation symmetry of a model composed of $n$ copies or "replicas" of a given theory, characterized by the exchange relations

$$\mathcal{T}(x)\mathcal{O}_i(y) = \mathcal{O}_{i+1}(y)\mathcal{T}(x) \quad \text{for} \quad y^1 > x^1, \tag{17}$$
$$= \mathcal{O}_i(y)\mathcal{T}(x) \quad \text{for} \quad x^1 > y^1, \tag{18}$$

where $\mathcal{O}_i(y)$ is any local field on copy number $i$, and with $\mathcal{O}_{n+1}(y) = \mathcal{O}_1(y)$.

The idea of quantum fields associated with branch points of Riemann surfaces in the context of entanglement appeared first in [32]. The general picture of branch point twist fields as symmetry fields associated to cyclic permutation symmetry of the $n$ Riemann surface's sheets, as per (17), was given in [24], where they were studied in massive IQFT. This description is however independent of integrability, and it was first used in massive QFT outside of integrability in [33].

Cyclic permutation symmetry is not naturally present in most IQFTs, but can be "manufactured" by considering a replica model, composed of $n$ copies of the original QFT (e.g. the sine-Gordon model). The connection to replica theories and multi-sheeted Riemann surfaces arises from the explicit formulae for entanglement measures, which generally depend on the quantity $\text{Tr}_A(\rho_A^n)$ where $\rho_A$ is the reduced density matrix associated to a particular region $A$ of the system. It is possible to show that the quantity $\text{Tr}_A(\rho_A^n)$ is proportional to a correlation function of branch point twist fields involving as many twist field insertions as boundary points between the region $A$ and the rest of the system. We will see an application of these ideas in Section 7 where we discuss the application of our results to the computation of the entanglement dynamics.

### 3.1 Form Factors and Form Factor Equations

Starting with the exchange relations (17), in IQFT one can formulate twist field form factor equations which generalize the standard form factor programme for local fields [3, 34]. These

equations were first given in [24] for diagonal theories and then in [27] for non-diagonal ones. They have been generalized to symmetry resolved branch point twist fields in [25, 26]. We will not review all these equations and their properties here but only those relations that are repeatedly used in the current paper, in particular the equations for one- and two-particle form factors. Let us start by defining

$$F_{a_1 \ldots a_k}(\theta_1, \cdots, \theta_k; \xi, n) := {}_n\langle 0| \mathcal{T}(0) |\theta_1, \cdots, \theta_k\rangle_{a_1 \ldots a_k; n}, \tag{19}$$

to be a $k$-particle form factor, that is, a matrix element of the field between the vacuum state and a $k$-particle state. Here ${}_n\langle 0|$ represents the vacuum state and $|\theta_1, \cdots, \theta_k\rangle_{a_1 \ldots a_k; n}$ represents an in-state of $k$ particles with rapidities $\theta_1, \ldots, \theta_k$ and quantum numbers $a_1 \ldots a_k$, both in the replica model. These quantum numbers generally contain two indices, one for the particle type and one for the copy number. However, in our computations we will generally restrict ourselves to a single copy and will therefore drop the copy index. This is because form factors of other copies can be obtained from these solutions by repeated use of the form factor equations.

The branch point twist field is a neutral field in relation to the sine-Gordon $U(1)$-symmetry that exchanges soliton and anti-soliton. This implies the vanishing of any twist-field form factors involving a different number of solitons and anti-solitons. At the one and two-particle level this means that

$$F_{ss}(\theta; \xi, n) = F_{\bar{s}\bar{s}}(\theta; \xi, n) = F_{\bar{s}b_k}(\theta, \xi; n) = F_{sb_k}(\theta, \xi; n) = F_s(\xi, n) = F_{\bar{s}}(\xi, n) = 0, \quad \forall \quad k \in \mathbb{Z}^+ \tag{20}$$

Here, we have used relativistic invariance and spinlessness of the twist field, which imply that the two-particle form factor depends on a single rapidity variable (the rapidity difference of the particles) and the one-particle form factor is rapidity independent. In addition, because of $\mathbb{Z}_2$ symmetry we also have

$$F_{b_{2k}b_{2p-1}}(\theta; \xi, n) = F_{b_{2k-1}}(\xi, n) = 0. \qquad \forall \qquad k, p \in \mathbb{Z}^+. \tag{21}$$

Under these considerations, Watson's equations for non-vanishing two-particle form factors and particles in the same copy can be summarized as

$$F_{s\bar{s}}(\theta; \xi, n) = S_+(\theta) F_{s\bar{s}}(-\theta; \xi, n) = F_{s\bar{s}}(2\pi i n - \theta; \xi, n), \tag{22}$$

$$F_{b_i b_j}(\theta; \xi, n) = S_{b_i b_j}(\theta) F_{b_i b_j}(-\theta; \xi, n) = F_{b_i b_j}(2\pi i n - \theta; \xi, n) \quad \text{for} \quad i - j \in 2\mathbb{Z}, \tag{23}$$

whereas the kinematic residue equations are

$$-i \operatorname*{Res}_{\theta = i\pi} F_{s\bar{s}}(\theta; \xi, n) = -i \operatorname*{Res}_{\theta = i\pi} F_{b_i b_i}(\theta; \xi, n) = \langle \mathcal{T} \rangle \quad \forall \quad i \in \mathbb{N}. \tag{24}$$

where $\langle \mathcal{T} \rangle$ is the vacuum expectation value of the branch point twist field in the ground state of the replica theory. Finally, the bound state residue equations are

$$-i \operatorname*{Res}_{\theta = i\pi u_{s\bar{s}}^c} F_{s\bar{s}}(\theta; \xi, n) = \Gamma_{s\bar{s}}^c F_c(\xi; n), \tag{25}$$

where $c$ is any particle that is formed as a bound state of $s + \bar{s}$ for rapidity difference $\theta = i\pi u_{s\bar{s}}^c$. In the breather sector we will use the bound state residue equation extensively and repeatedly

to obtain lower particle form factors of heavier breathers in a process known as "fusion". For this reason it is convenient to write the more general equation

$$-i\operatorname*{Res}_{\theta=\theta_0} F_{b_i b_j a_1 \ldots a_k}(\theta + iu, \theta_0 - i\tilde{u}, \theta_1 \cdots, \theta_k; \xi, n) = \Gamma^{b_{i+j}}_{b_i b_j} F_{b_{i+j} a_1 \ldots a_k}(\theta, \theta_1 \cdots, \theta_k; \xi, n), \qquad (26)$$

where $a_1, \ldots, a_k$ are any particle combination for which the form factor is non-vanishing and $u + \tilde{u} = u^{i+j}_{ij}$ where $\theta = i\pi u^{i+j}_{ij}$ is the pole of the scattering matrix $S_{b_i b_j}(\theta)$ corresponding to the formation of breather $b_{i+j}$. Similarly, $u$ and $\tilde{u}$ are related to the poles of $S_{b_j b_{i+j}}(\theta)$ and $S_{b_i b_{i+j}}(\theta)$.

## 4   Soliton-Antisoliton Form Factors

In the following we summarise the necessary formulas for the two-particle soliton-antisoliton form factors of the branch point twist field. Although these quantities were already derived in [27], the formulas were strictly speaking only justified in the repulsive regime of the sine-Gordon model. As we show below they are, nevertheless, valid in the attractive regime as well once a proper analytic continuation in the parameter $\xi$ is considered. Let us first discuss the minimal form factor of these objects, which we denote by $G(\theta; \xi, n)$. This is the "minimal solution" to using Eq. (22) which can be constructed in the manner shown in [24], which itself generalizes a standard method in the context of the form factor programme (see e.g. [35]). This method takes as starting point the $S$-matrix involved in the middle identity ($S_+(\theta)$ in the present case) of (22) and assumes that it admits a representation of the type

$$S(\theta) = \exp\left[\int_0^\infty \frac{\mathrm{d}t}{t} g(t) \sinh \frac{t\theta}{i\pi}\right], \qquad (27)$$

for some function $g(t)$. If such a representation exists, then a minimal solution the equation (22) is given by

$$f(\theta) = \mathcal{N} \exp\left[\int_0^\infty \frac{\mathrm{d}t}{t} \frac{g(t)}{\sinh nt} \sin^2\left(\frac{itn}{2}\left(1 + \frac{i\theta}{\pi}\right)\right)\right]. \qquad (28)$$

where $\mathcal{N}$ is a normalization constant. To obtain the minimal form factor $G(\theta; \xi, n)$ of interest, we therefore need to write $S_+$ in the form (27). This is straightforward since

$$S_+(\theta) = \left(\frac{\sinh \frac{\theta}{\xi}}{\sinh \frac{i\pi - \theta}{\xi}} + \frac{\sinh \frac{i\pi}{\xi}}{\sinh \frac{i\pi - \theta}{\xi}}\right) S^{ss}_{ss}(\theta) := s(\theta) S^{ss}_{ss}(\theta), \qquad (29)$$

with

$$s(\theta) := \frac{\sin \frac{\pi - i\theta}{2\xi}}{\sin \frac{\pi + i\theta}{2\xi}}. \qquad (30)$$

The function $S^{ss}_{ss}(\theta)$ already has an exponential representation (5) and one can easily write a similar representation for the function $s(\theta)$ as well

$$s(\theta) = \exp\left[-2\int_0^\infty \frac{\mathrm{d}t}{t} \frac{\sinh((\xi - 1)t)\sinh \frac{it\theta}{\pi}}{\sinh \xi t}\right]. \qquad (31)$$

An important remark is that the integral (5) is convergent for any $0 < \xi < 1$. Nevertheless (31) is only convergent for $\frac{1}{2} < \xi < 1$. To be precise, for other values of $\xi$ an alternative representation of the function above has to be used given by

$$s(\theta) = \exp\left[2\int_0^\infty \frac{dt}{t} \frac{\sinh\left(((2p+1)\xi - 1)\,t\right)\sinh\frac{t\theta}{i\pi}}{\sinh\xi t}\right] \qquad \text{for} \qquad \frac{1}{2p} \geqslant \xi > \frac{1}{2p+2} \qquad (32)$$

with $p \in \mathbb{Z}^+$. Thus, we have two different representations of the minimal form factor $G(\theta; \xi, n)$ depending on whether or not $\xi - \frac{1}{2} > 0$ or $\xi - \frac{1}{2} \leqslant 0$ which we denote by $G_\pm(\theta; \xi, n)$, respectively. Interestingly, the value $\xi = \frac{1}{2}$ is precisely the threshold for the formation of breathers and this is no coincidence. From the symmetry arguments presented in subsection 3.1 we know that the branch point twist field has vanishing one-particle form factors for odd-indexed breathers. However, the presence of non-zero one-particle breather form factors for even indices is allowed as we show later. This means that the two-particle soliton-antisoliton form factor of the branch-point twist field must have bound state poles at imaginary rapidity values $\theta = i\pi(1 - 2k\xi)$ for $k = 1, \ldots, [\frac{1}{2\xi}]$. Equivalently, we can formulate this statement as the dynamical pole axiom (25) which we now specialise to even breather bound states

$$-i\operatorname*{Res}_{\theta = i\pi(1-2k\xi)} F_{s\bar{s}}(\theta; \xi, n) = \Gamma_{s\bar{s}}^{b_{2k}} F_{b_{2k}}(\xi, n)\,. \qquad (33)$$

Notice that each new representation of $s(\theta)$ from (32), corresponds to a new breather with an even index entering the spectrum of the theory.

Thus, when writing down the minimal part of $F_{s\bar{s}}(\theta; \xi, n)$ we have two alternative representations: if we employ the $S$-matrix representation (32) together with (5) and apply the standard machinery (28) to obtain the minimal form factor, the result possesses no breather bound state poles. This feature, is generally what is meant by "minimal solution". In this case the dynamical pole equation (33) can only be satisfied by multiplying the minimal form factor with another function which incorporates the required poles, similarly as for kinematic poles [24]. On the other hand, if one uses the analytically continued solution (31) instead of (32), the dynamical pole axiom (33) is automatically satisfied by the minimal form factor. In other words, this form factor is no-longer "minimal" in the standard sense, but includes also poles in the physical sheet corresponding to bound states.

Let us now continue our derivation for the minimal form factor, where the above discussed features can be explicitly demonstrated. The minimal form factor can be written as

$$G(\theta; \xi, n) = \varphi(\theta; \xi, n)\Phi(\theta; \xi, n)\,, \qquad (34)$$

where the function $\Phi(\theta; \xi, n)$ follows from the integral representation of $S_{ss}^{ss}(\theta)$ and can be written as

$$\Phi(\theta; \xi, n) = -i\sinh\frac{\theta}{2n}\exp\left[\int_0^\infty \frac{dt}{t} \frac{\sinh\left(\frac{1}{2}(\xi-1)t\right)\sinh^2\left(\frac{t}{2}\left(n - \frac{\theta}{i\pi}\right)\right)}{\cosh\frac{t}{2}\sinh\frac{\xi t}{2}\sinh nt}\right]\,, \qquad (35)$$

or, alternatively, as an infinite product of Gamma functions:

$$\Phi(\theta;\xi,n) = -i\sinh\frac{\theta}{2n}\prod_{k,p=0}^{\infty}\left[\frac{\Gamma\left(\frac{p+n+(k+1)\xi}{2n}\right)^2\Gamma\left(1+\frac{\frac{i\theta}{\pi}+p+1+k\xi}{2n}\right)\Gamma\left(\frac{-\frac{i\theta}{\pi}+p+1+k\xi}{2n}\right)}{\Gamma\left(\frac{p+n+k\xi+1}{2n}\right)^2\Gamma\left(1+\frac{\frac{i\theta}{\pi}+p+(k+1)\xi}{2n}\right)\Gamma\left(\frac{-\frac{i\theta}{\pi}+p+(k+1)\xi}{2n}\right)}\right]^{(-1)^p}$$

(36)

However, from a numerical viewpoint, the most useful representation is mixed, combining both a finite product of Gamma-fuctions and an integral part. This representation (86) is given in Appendix A. This kind of mixed representation was first used in [22] and is very rapidly convergent.

The function $\varphi(\theta;\xi,n)$ in (34) follows from either the representation (31) valid for $\xi > \frac{1}{2}$

$$\varphi_+(\theta;\xi,n) = \exp\left[-2\int_0^{\infty}\frac{dt}{t}\frac{\sinh\left((\xi-1)t\right)\sinh^2\left(\frac{t}{2}\left(n-\frac{\theta}{i\pi}\right)\right)}{\sinh(nt)\sinh(\xi t)}\right]$$

$$= \prod_{k=0}^{\infty}\frac{\Gamma\left(\frac{n+2k\xi+1}{2n}\right)^2\Gamma\left(\frac{-\frac{i\theta}{\pi}+2\xi(k+1)-1}{2n}\right)\Gamma\left(1+\frac{\frac{i\theta}{\pi}+2\xi(k+1)-1}{2n}\right)}{\Gamma\left(\frac{n+2\xi(k+1)-1}{2n}\right)^2\Gamma\left(\frac{-\frac{i\theta}{\pi}+2k\xi+1}{2n}\right)\Gamma\left(1+\frac{\frac{i\theta}{\pi}+2k\xi+1}{2n}\right)},$$

(37)

or from (32), which is instead valid for $\frac{1}{2p} \geqslant \xi > \frac{1}{2p+2}$ and $p \in \mathbb{Z}^+$

$$\varphi_-(\theta;\xi,n) = \exp\left[-2\int_0^{\infty}\frac{dt}{t}\frac{\sinh\left(((2p+1)\xi-1)t\right)\sinh^2\left(\frac{t}{2}\left(n-\frac{\theta}{i\pi}\right)\right)}{\sinh(nt)\sinh(\xi t)}\right]$$

$$= \prod_{k=0}^{\infty}\frac{\Gamma\left(\frac{n+2(k-p)\xi+1}{2n}\right)^2\Gamma\left(\frac{-\frac{i\theta}{\pi}+2(k+p+1)\xi-1}{2n}\right)\Gamma\left(1+\frac{\frac{i\theta}{\pi}+2(k+p+1)\xi-1}{2n}\right)}{\Gamma\left(\frac{n+2(k+p+1)\xi-1}{2n}\right)^2\Gamma\left(\frac{-\frac{i\theta}{\pi}+2(k-p)\xi+1}{2n}\right)\Gamma\left(1+\frac{\frac{i\theta}{\pi}+2(k-p)\xi+1}{2n}\right)},$$

(38)

As before, we can also write a mixed representations (see Eq. (87) and (88)). Similar to the discussion following (31)-(31), the minimal form factors

$$G_\pm(\theta;\xi,n) = \varphi_\pm(\theta;\xi,n)\Phi(\theta;\xi,n),$$

(39)

are two representations both satisfy Eq. (22), but whereas $G_+(\theta;\xi,n)$ includes bound state poles at $\theta = i\pi(1-2k\xi)$ for $k = 1,\ldots,[\frac{1}{2\xi}]$, $G_-(\theta;\xi,n)$ does not. Instead the necessary bound state poles can be introduced by simply dividing $G_-(\theta;\xi,n)$ by standard CDD factors of the type

$$\prod_{k=1}^{[\frac{1}{2\xi}]}\left(\cosh\frac{\theta}{n}-\cos\frac{\pi(1-2k\xi)}{n}\right).$$

(40)

A rigorous demonstration of this fact is presented in Appendix D. In this Appendix, the fulfilment of (25) with our soliton and breather form factors is numerically checked as well, and

we also derive some identities involving fractions of the minimal soliton-antisoliton form factors $G_\pm(\theta; \xi, n)$ and breather form factor $R(\theta; \xi, n)$ (derived in the next section) based on (25).

Now that we have found a minimal form factor that incorporates also the bound state poles, we just need to introduce the kinematic pole that ensures our solution satisfies (24). This kinematic pole can be introduced by multiplying with a function already presented in [24]. The final formulae for particles on the same copy are

$$
\begin{aligned}
F_{s\bar{s}}(\theta; \xi, n) &= \frac{\langle \mathcal{T} \rangle \sin \frac{\pi}{n}}{2n \sinh \frac{i\pi - \theta}{2n} \sinh \frac{i\pi + \theta}{2n}} \frac{G_+(\theta; \xi, n)}{G_+(i\pi; \xi, n)} \\
&= \frac{\langle \mathcal{T} \rangle \sin \frac{\pi}{n}}{2n \sinh \frac{i\pi - \theta}{2n} \sinh \frac{i\pi + \theta}{2n}} \left[ \prod_{k=1}^{[\frac{1}{2\xi}]} \frac{\cos \frac{\pi}{n} - \cos \frac{\pi(1-2k\xi)}{n}}{\cosh \frac{\theta}{n} - \cos \frac{\pi(1-2k\xi)}{n}} \right] \frac{G_-(\theta; \xi, n)}{G_-(i\pi; \xi, n)} \, .
\end{aligned}
\tag{41}
$$

We stress again that the two formulas are completely identical on the physical sheet and that the first line is the same expression derived for the repulsive regime in [24].

## 5   Breather Form Factors

In this section we focus on the breather sector of the theory, where the $S$-matrices are diagonal. The form factors

$$
F_{b_1 b_1}(\theta; \xi, n), \qquad F_{b_1 b_1 b_1 b_1}(\theta_1, \theta_2, \theta_3, \theta_4; \xi, n) \, ,
\tag{42}
$$

can be easily obtained from known results for the sinh-Gordon model under the replacement $B = -2\xi$. With this identification one can then take the form factor solutions found in [24, 28] and employ fusion to construct the chains of form factors

$$
\begin{aligned}
F_{b_1 b_1 b_1 b_1}(\theta_1, \theta_2, \theta_3, \theta_4; \xi, n) &\mapsto F_{b_2 b_1 b_1}(\theta_1, \theta_2, \theta_3; \xi, n) \\
&\mapsto F_{b_2 b_2}(\theta; \xi, n) \quad \text{or} \quad F_{b_3 b_1}(\theta; \xi, n) \mapsto F_{b_4}(\xi, n) \, .
\end{aligned}
\tag{43}
$$

and

$$
F_{b_1 b_1}(\theta; \xi, n) \mapsto F_{b_2}(\xi, n) \, .
\tag{44}
$$

A nice example of this approach was given in Appendix A of [36] for the form factors of exponential fields.

### 5.1   Minimal Form Factor and Form Factors of $b_1$

Although we take the sinh-Gordon solutions as starting point, it is still useful to say a few words about the basic structure of those solutions, specially the minimal form factor. This function provides a minimal solution to the equations (23) for $i = j = 1$ and two breathers in the same copy. It can be easily adapted from the solutions presented in various papers [4,10,20–22] and the techniques for the computation of minimal form factors introduced in [24]. The generalization

to branch point twist fields of the representation given in [10] takes the form

$$R(\theta;\xi,n) = \exp\left[2\int_0^\infty \frac{dt}{t}\frac{\sinh\frac{\xi t}{2}\sinh\frac{t(1+\xi)}{2}\cosh\left(t\left(n+\frac{i\theta}{\pi}\right)\right)}{\cosh\frac{t}{2}\sinh(nt)}\right]$$

$$=\prod_{k=0}^{\infty}\left[\frac{\Gamma\left(\frac{-\frac{i\theta}{\pi}-\xi+k}{2n}\right)\Gamma\left(1+\frac{\frac{i\theta}{\pi}-\xi+k}{2n}\right)\Gamma\left(\frac{-\frac{i\theta}{\pi}+1+\xi+k}{2n}\right)\Gamma\left(1+\frac{\frac{i\theta}{\pi}+1+\xi+k}{2n}\right)}{\Gamma\left(\frac{-\frac{i\theta}{\pi}+k}{2n}\right)\Gamma\left(1+\frac{\frac{i\theta}{\pi}+k}{2n}\right)\Gamma\left(\frac{-\frac{i\theta}{\pi}+k+1}{2n}\right)\Gamma\left(1+\frac{\frac{i\theta}{\pi}+k+1}{2n}\right)}\right]^{(-1)^k} \quad (45)$$

This function has the useful properties:

$$\lim_{\theta\to\infty} R(\theta;\xi,n)=1 \quad \text{and} \quad R(0;\xi,n)=0\,. \quad (46)$$

A similar discussion as presented in the previous section also applies to this solution. First, although $R(\theta;\xi,n)$ is constructed from the sinh-Gordon minimal form factor, it has very different analytic properties. Indeed, once more $R(\theta;\xi,n)$ is not minimal, in the strictest sense of having no poles in the physical sheet. $R(\theta;\xi,n)$ does have poles in the physical sheet, when the coupling allows for the the first breather to form higher breather bound states. Therefore, the solution (47) is valid for all values of the coupling $\xi$, with the function $R(\theta;\xi,n)$ introducing bound state poles as needed. Second, the formula is once more only convergent for $\xi>\frac{1}{2}$ and this can be numerically addressed by employing the mixed representation (89).

The full two-particle form factor is then given by

$$F_{b_1 b_1}(\theta;\xi,n) = \frac{\langle\mathcal{T}\rangle\sin\frac{\pi}{n}}{2n\,\sinh\frac{i\pi-\theta}{2n}\sinh\frac{i\pi+\theta}{2n}}\frac{R(\theta;\xi,n)}{R(i\pi;\xi,n)}\,, \quad (47)$$

The four-particle form factor can be read off from [28] and takes the form

$$F_{b_1 b_1 b_1 b_1}(\theta_1,\theta_2,\theta_3,\theta_4;\xi,n) = H(\xi,n)Q(x_1,x_2,x_3,x_4;\xi,n)\prod_{1\leqslant i<j\leqslant 4}\frac{R(\theta_i-\theta_j;\xi,n)}{(x_i-\omega x_j)(x_j-\omega x_i)}\,, \quad (48)$$

with

$$H(\xi,n)=\langle\mathcal{T}\rangle\frac{4\omega^6\sin^2\frac{\pi}{n}}{n^2 R(i\pi;\xi,n)^2}\,, \qquad x_i=e^{\frac{\theta_i}{n}}\,, \qquad \omega=e^{\frac{i\pi}{n}}\,. \quad (49)$$

and

$$\begin{aligned}
Q(x_1,x_2,x_3,x_4;\xi,n) = {} & \sigma_4\big[\sigma_2^4+q_1(\xi,n)\sigma_2(\sigma_3^2+\sigma_1^2\sigma_4)+q_2(\xi,n)\sigma_1\sigma_2^2\sigma_3+q_3(\xi,n)\sigma_1^2\sigma_3^2\\
& q_4(\xi,n)\sigma_2^2\sigma_4+q_5(\xi,n)\sigma_1\sigma_3\sigma_4+q_6(\xi,n)\sigma_4^2\big]\,.
\end{aligned} \quad (50)$$

Here $\sigma_i$ are the elementary symmetric polynomials on variables $\{x_1,x_2,x_3,x_4\}$ and the coefficients $q_i(\xi,n)$ where given in the Appendix of [28] (which unfortunately contains a typo). Calling

$$c(a):=\cos\frac{\pi a}{2n}\,. \quad (51)$$

they can be rewritten as

$$
\begin{aligned}
q_1(\xi, n) &= c(1)^{-1} \left(1 + 2c(2)\right) \left(c(3) - c(1 + 2\xi)\right), \\
q_2(\xi, n) &= -c(1)^{-1} \left(c(2\xi + 1) + 4c(1) + c(3)\right), \\
q_3(\xi, n) &= 2c(2(1 + \xi)) + 2c(2\xi) + 2c(2) + 3, \\
q_4(\xi, n) &= 2 \left(3c(2\xi) + 3c(2(1 + \xi)) + c(2(2 + \xi)) + c(2(1 - \xi)) + c(2(1 + 2\xi))\right. \\
&\quad \left. + 3c(2) - c(4) + 1\right), \\
q_5(\xi, n) &= -2 \left(6 + 6c(2) + 4c(4) + c(6) + c(2(2 - \xi)) + 5c(2\xi) + c(4\xi) + 5c(2(1 + \xi))\right. \\
&\quad \left. + c(4(1 + \xi)) + 2c(2(2 + \xi)) + c(2(3 + \xi)) + 2c(2(1 - \xi)) + c(2(1 + 2\xi))\right), \\
q_6(\xi, n) &= 8c(2)^2 \left(3 + 3c(2) - c(4) + 3c(2\xi) + 3c(2(1 + \xi)) + c(4(1 + \xi)) + c(2(2 + \xi))\right. \\
&\quad \left. + c(2(1 - \xi)) + c(2(1 + 2\xi))\right),
\end{aligned}
\tag{52}
$$

## 5.2 Fusion Procedure

In this section we present the results of the fusion procedure as described schematically in (43). The simplest form factor to be obtained from the bootstrap approach outlined before is $F_{b_2}(\xi, n)$. The breather $b_2$ is a bound state of two $b_1$ breathers corresponding to the simple pole of $S_{b_1 b_2}(\theta)$ at $\theta = i\pi\xi$. The bound state residue equation simply tells us that

$$
-i \operatorname*{Res}_{\theta = i\pi\xi} F_{b_1 b_1}(\theta; \xi, n) = \Gamma^{b_2}_{b_1 b_1} F_{b_2}(\xi, n),
\tag{53}
$$

We also know that the minimal form factor $R(\theta; \xi, n)$ satisfies the equation

$$
R(\theta; \xi, n) = S_{b_1 b_1}(\theta) R(-\theta; \xi, n),
\tag{54}
$$

and so, at the pole we have that

$$
-i \operatorname*{Res}_{\theta = i\pi\xi} R(\theta; \xi, n) = -i \operatorname*{Res}_{\theta = i\pi\xi} S_{b_1 b_1}(\theta) R(-\theta; \xi, n) = (\Gamma^{b_2}_{b_1 b_1})^2 R(-i\pi\xi; \xi, n).
\tag{55}
$$

Putting all factors together, this gives the formula

$$
F_{b_2}(\xi, n) = \frac{\langle \mathcal{T} \rangle \sin \frac{\pi}{n} \sqrt{2 \tan \pi\xi}}{2n \, \sinh \frac{i\pi(1-\xi)}{2n} \, \sinh \frac{i\pi(1+\xi)}{2n}} \frac{R(-i\pi\xi; \xi, n)}{R(i\pi; \xi, n)},
\tag{56}
$$

For $n \to 1$ the form factor vanishes as expected (since the twist field becomes the identity if the replica number is 1). However the limit

$$
\lim_{n \to 1} \frac{F_{b_2}(\xi, n)}{1 - n} = \frac{\pi \sqrt{\tan \pi\xi}}{\sqrt{2} \cos^2 \frac{\pi\xi}{2}} \frac{R(-i\pi\xi; \xi, 1)}{R(i\pi; \xi, 1)},
\tag{57}
$$

is non-zero. This limit plays a role in computations of the von Neumann entropy.

Note that the breather $b_2$ is only present for $\xi < \frac{1}{2}$. Fig. 1 shows the function (56) for several choices of $\xi$ and $n$.

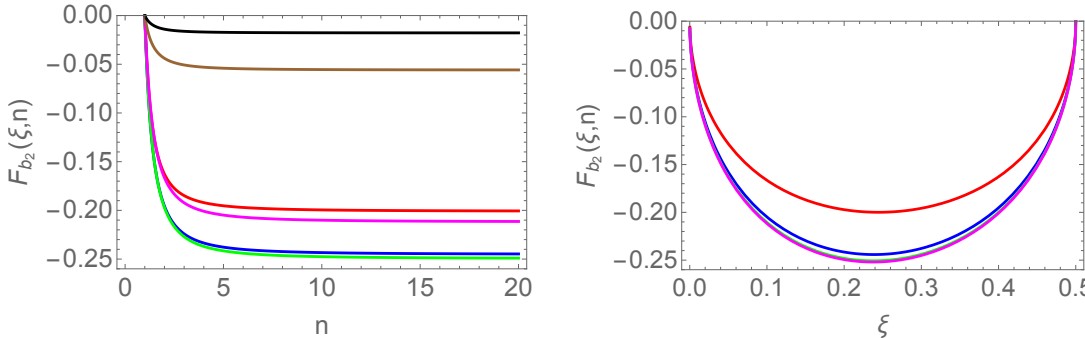

Figure 1: Left: The one-particle form factor $F_{b_2}(\xi, n)$ as a function of $n$ for $\xi = 0.4$ (pink), 0.3 (blue), 0.2 (green), 0.1 (red), 0.05 (brown) and 0.005 (black). Right: The one-particle form factor $F_{b_2}(\xi, n)$ as a function of $\xi$ for $n = 2$ (red), 5 (blue), 12 (green), 50 (magenta).

### 5.2.1 Higher Breather Form Factors

Let us now consider a more involved fusion-based computation, namely that giving the form factor $F_{b_2 b_1 b_1}(\theta_1, \theta_2, \theta_3; \xi, n)$ from the four-particle form factor (48). The key equation in this case is

$$-i \operatorname*{Res}_{\theta=\theta_1} F_{b_1 b_1 b_1 b_1}\left(\theta + \frac{i\pi\xi}{2}, \theta_1 - \frac{i\pi\xi}{2}, \theta_2, \theta_3; \xi, n\right) = \Gamma_{b_1 b_1}^{b_2} F_{b_2 b_1 b_1}(\theta_1, \theta_2, \theta_3; \xi, n), \quad (58)$$

Considering the formula (48) we see once more that the pole will originate from of the $R$-factors in the product, giving the contribution (55). More precisely, we obtain a solution of the form

$$F_{b_2 b_1 b_1}(\theta_1, \theta_2, \theta_3; \xi, n) = H_{211}(\xi, n) Q_{211}(x_1, x_2, x_3; \xi, n)$$
$$\times \frac{R(\theta_{23}; \xi, n) R(\theta_{12} + \frac{i\pi\xi}{2}; \xi, n) R(\theta_{13} + \frac{i\pi\xi}{2}; \xi, n) R(\theta_{12} - \frac{i\pi\xi}{2}; \xi, n) R(\theta_{13} - \frac{i\pi\xi}{2}; \xi, n)}{(x_2 - \omega x_3)(x_3 - \omega x_2)(x_1 - x_2 \omega\sqrt{\beta})(x_2 - x_1 \omega\sqrt{\beta})(x_1 - x_3 \omega\sqrt{\beta})(x_3 - x_1 \omega\sqrt{\beta})}, \quad (59)$$

where $\beta = e^{-\frac{i\pi\xi}{n}}$ and $Q_{211}(x_1, x_2, x_3; \xi, n)$ is obtained from evaluating $Q(x_1 \beta^{-\frac{1}{2}}, x_1 \beta^{\frac{1}{2}}, x_2, x_3)$ which simplifies with part of the denominator of (48) giving

$$\begin{aligned}
Q_{211}(x_1, x_2, x_3; \xi, n) &= \sigma_2 \big[ (\sigma_2^2 + \hat{\sigma}_1^2 \sigma_1^2 + \hat{\sigma}_1^4) c(1) + 2\sigma_2 \hat{\sigma}_1^2 c(\xi) c(\xi - 5) \\
&\quad - 2\sigma_1 \hat{\sigma}_1 (\sigma_2 + \hat{\sigma}_1^2) c(\xi + 2) c(2\xi - 1) \\
&\quad + 2\sigma_2 \hat{\sigma}_1^2 (c(1) c(2(\xi + 2)) - c(\xi) c(3\xi + 1)) \\
&\quad + 2(\sigma_1^2 - \sigma_2) \hat{\sigma}_1^2 c(\xi) (c(3\xi - 1) - c(3 - \xi)) \big],
\end{aligned} \quad (60)$$

and $\sigma_1 = x_2 + x_3$, $\sigma_2 = x_2 x_3$ and $\hat{\sigma}_1 = x_1$. As for the constant, we obtain

$$H_{211}(\xi, n) = \langle \mathcal{T} \rangle \frac{2\omega^3 \beta \sin \frac{\pi}{2n} \sin \frac{\pi}{n} \Gamma_{b_1 b_1}^{b_2}}{n^2 \sin \frac{\pi(\xi+1)}{2n} \sin \frac{\pi(\xi-1)}{2n}} \frac{R(-i\pi\xi; \xi, n)}{R(i\pi; \xi, n)^2} = \frac{4\omega^3 \beta \sin \frac{\pi}{2n} F_{b_2}(\xi, n)}{n R(i\pi; \xi, n)}. \quad (61)$$

Having now seen two applications of the fusion procedure it is easy to proceed for other form factors. We present more details of those computations in Appendix B. Here we just summarize the main formulae:

$$F_{b_3 b_1}(\theta_{12}; \xi, n) = H_{31}(\xi, n) Q_{31}(x_1, x_2; \xi, n) \frac{R(\theta_{12}; \xi, n) R(\theta_{12} + i\pi\xi; \xi, n) R(\theta_{12} - i\pi\xi; \xi, n)}{(x_1 - x_2 \omega\beta)(x_2 - x_1 \omega\beta)(x_1 \alpha - x_2)(x_2 \alpha - x_1 \beta)}. \quad (62)$$

with $H_{31}(\xi, n)$ and $Q_{31}(x_1, x_2; \xi, n)$ given in (93), (92), respectively.

$$F_{b_2 b_2}(\theta_{12}; \xi, n) = H_{22}(\xi, n) Q_{22}(x_1, x_2; \xi, n) \frac{R(\theta_{12}; \xi, n)^2 R(\theta_{12} + i\pi\xi; \xi, n) R(\theta_{12} - i\pi\xi; \xi, n)}{(x_1 - \alpha x_2)(x_2 - \alpha x_1)(x_1 - \alpha\beta x_2)(x_2 - \alpha\beta x_1)}, \quad (63)$$

with $H_{22}(\xi, n)$ and $Q_{22}(x_1, x_2; \xi, n)$ given by (104) and (102) and, finally

$$\begin{aligned} F_{b_4}(\xi, n) &= \langle \mathcal{T} \rangle \frac{\sin\frac{\pi}{n} \sin\frac{\pi}{2n}(1 + 2\cos\frac{\pi\xi}{n})\cos\frac{\pi(1-\xi)}{2n} \Gamma^{b_4}_{b_3 b_1} \Gamma^{b_3}_{b_2 b_1} \Gamma^{b_2}_{b_1 b_1}}{2n^2 \sin^2\frac{\pi(1+\xi)}{2n} \sin\frac{\pi(1-2\xi)}{2n} \sin\frac{\pi(1-3\xi)}{2n}} \\ &\times \frac{R(-3\pi i\xi; \xi, n) R(-2\pi i\xi; \xi, n)^2 R(-i\pi\xi; \xi, n)^3}{R(i\pi; \xi, n)^2}, \end{aligned} \quad (64)$$

which, as we see in Appendix B can be obtained from either fusing $b_3$ and $b_1$ or $b_2$ with itself, giving identical results. Before ending this section, it is worth noting that Watson's equations and the bound state residue equation for form factors can be repeatedly used to obtain the form factors of breather $b_k$ starting with a form factor involving $k$ breathers of type $b_1$ in a more systematic manner. This technique is described for instance in equation (A.3) of Appendix A in [37]. This method would allow us for instance to reduce (48) to (64) by simultaneously fusing all particles. The result is the same as presented here.

## 5.3 Some Consistency Checks

Apart from the $\Delta$ sum rule that we will discuss later, there are a few properties that the form factors must satisfy and which help us make sure these formulae are correct. One of the strongest tests is the clustering decomposition property which states that in the absence of internal symmetries, form factors factorize into products of lower particle number form factors if a subset of the rapidities is sent to infinity. More precisely, for the form factors above we expect that

$$\lim_{\theta \to \infty} F_{b_1 b_3}(\theta; \xi, n) = 0, \qquad \lim_{\theta \to \infty} F_{b_2 b_2}(\theta; \xi, n) = \frac{F_{b_2}(\xi, n)^2}{\langle \mathcal{T} \rangle}, \quad (65)$$

and

$$\lim_{\theta_1 \to \infty} F_{b_2 b_1 b_1}(\theta_1, \theta_2, \theta_3; \xi, n) = \frac{F_{b_2}(\xi, n) F_{b_1 b_1}(\theta_{23}; \xi, n)}{\langle \mathcal{T} \rangle}, \qquad \lim_{\theta_1, \theta_2 \to \infty} F_{b_2 b_1 b_1}(\theta_1, \theta_2, \theta_3; \xi, n) = 0. \quad (66)$$

These identities can be easily checked thanks to the first property in (46). The first property in (65) follows from observing that for $\theta_1 \to \infty$ the denominator of the form factor (62) scales

with $x_1^4$ whereas the numerator (that is, the function $Q_{31}(x_1, x_2, x_2; \xi, n)$) scales as $x_1^3$. A similar argument applies to the second equality in (66). The second identity in (65) follows from

$$\lim_{\theta_1 \to \infty} Q_{22}(x_1, x_2; \xi, n) \sim 2\omega\sqrt{\omega}\beta^2 c(1)x_1^4 , \tag{67}$$

and

$$\lim_{\theta_1 \to \infty} \frac{R(\theta_{12}; \xi, n)^2 R(\theta_{12} + i\pi\xi; \xi, n)R(\theta_{12} - i\pi\xi; \xi, n)}{(x_1 - \omega x_2)(x_2 - \omega x_1)(x_1 - \omega\beta x_2)(x_2 - \omega\beta x_1)} \sim \frac{1}{\omega^2 \beta x_1^4} \tag{68}$$

together with the formula (104). The first identity in (66) follows from

$$\lim_{\theta_1 \to \infty} Q_{211}(x_1, x_2, x_3; \xi, n) \sim c(1)x_1^4 x_2 x_3 , \tag{69}$$

$$\lim_{\theta_1 \to \infty} \frac{R(\theta_{23}; \xi, n)R(\theta_{12} + \frac{i\pi\xi}{2}; \xi, n)R(\theta_{13} + \frac{i\pi\xi}{2}; \xi, n)R(\theta_{12} - \frac{i\pi\xi}{2}; \xi, n)R(\theta_{13} - \frac{i\pi\xi}{2}; \xi, n)}{(x_2 - \omega x_3)(x_3 - \omega x_2)(x_1 - x_2\omega\sqrt{\beta})(x_2 - x_1\omega\sqrt{\beta})(x_1 - x_3\omega\sqrt{\beta})(x_3 - x_1\omega\sqrt{\beta})}$$
$$\sim \frac{R(\theta_{23}; \xi, n)}{\omega^2 \beta x_1^4 (x_2 - \omega x_3)(x_3 - \omega x_2)} \tag{70}$$

Comparing with (47) and (56) we find that the clustering property is exactly reproduced. We may also check that the solution $F_{b_2 b_2}(\theta; \xi, n)$ satisfies the kinematic residue equation (24) which indeed it does. This can be shown by employing the non-trivial identity

$$R(-i\pi\xi; \xi, n)^2 R(i\pi(1 - \xi); \xi, n)R(i\pi(1 + \xi); \xi, n) = \frac{n \tan \frac{\pi\xi}{2n} \sin \frac{\pi(1+\xi)}{2n} \sin \frac{\pi(\xi-1)}{2n}}{2\omega^4 \sin \frac{\pi}{2n} \sin \frac{\pi(1+2\xi)}{2n} \tan \pi\xi} , \tag{71}$$

which can be established with the help of the $\Gamma$-function representation given in Appendix A.

# 6  Consistency Checks by $\Delta$ Sum Rule

The $\Delta$ sum rule [38] is one of the most useful and common methods for testing form factor solutions. It gives a relationship between the conformal dimension of a local field $\mathcal{T}$ and a certain integral involving the two point function $_n\langle 0|\mathcal{T}(0)\Theta(r)|0\rangle_n$ where $\Theta$ is the trace of the stress-energy tensor and $|0\rangle_n$ is again the vacuum state in the replica theory. In its integrated form given for instance in [39] and after generalizing to branch point twist fields, the rule can be expressed as follows:

$$\Delta_{\mathcal{T}} = -\frac{n}{2\langle\mathcal{T}\rangle} \sum_{k=1}^{\infty} \sum_{a_1 \dots a_k} \int_{-\infty}^{\infty} \frac{d\theta_1 \dots d\theta_k}{k!(2\pi)^k} \frac{F_{a_1 \dots a_k}^{\mathcal{T}}(\theta_1, \dots, \theta_k; \xi, n)F_{a_1 \dots a_k}^{\Theta}(\theta_1, \dots, \theta_k; \xi)^*}{\left(\sum_{p=1}^k m_p \cosh\theta_p\right)^2} , \tag{72}$$

where $\Delta_{\mathcal{T}} = \frac{c}{24}(n - \frac{1}{n})$ is the conformal dimension of the branch point twist field [24, 32, 40] and we have now added a superindex to the form factors to indicate the quantum field they correspond to. The second sum is over all possible choices of particle types $a_p$ with masses $m_p$.

As usual with this type of expansion, convergence of the sum is expected to be quick, and the main contributions come from the one- and two-particle form factors. Hence, if we can show such near saturation we can be confident that our form factors solutions are correct.

Let $\Delta_{\mathcal{T}}^{(\ell(\xi))}$ be the conformal dimension of the branch point twist field as given by (72) in the regime where $\ell(\xi)$ breathers are present. Although the exact value of $\Delta_{\mathcal{T}}$ is independent of $\xi$ the number and contribution of the terms in the sum changes substantially depending on the coupling. In what follows we present numerical results for the sum above for $\ell(\xi) = 1, 2, 3$ and 4. For this we need first to obtain the one- and two-particle breather form factors of the stress-energy tensor in the sine-Gordon model. This can be done in a similar fashion as for the branch point twist fields, namely starting from the sinh-Gordon solutions presented in [22] and carrying out the fusion procedure. The results are presented in appendix E.

| $n$ | $\Delta_{\mathcal{T}}$ | $s\bar{s}$ | $b_1 b_1$ | $\Delta_{\mathcal{T}}^{(1)}$ | $n$ | $\Delta_{\mathcal{T}}$ | $s\bar{s}$ | $b_1 b_1$ | $\Delta_{\mathcal{T}}^{(1)}$ |
|---|---|---|---|---|---|---|---|---|---|
| 2 | 0.0625 | 0.0602025 | 0.0008771 | 0.0610796 | 2 | 0.0625 | 0.0618871 | 0.0000835 | 0.0619705 |
| 3 | 0.11111 | 0.1064464 | 0.0016783 | 0.1081246 | 3 | 0.11111 | 0.1098190 | 0.0001636 | 0.1099826 |
| 4 | 0.15625 | 0.1493874 | 0.0024134 | 0.1518008 | 4 | 0.15625 | 0.1543269 | 0.0002368 | 0.1545637 |
| 5 | 0.2 | 0.1910316 | 0.0031194 | 0.1941510 | 5 | 0.2 | 0.1974738 | 0.0003068 | 0.1977806 |

Table 1: The contributions to the sum (73) from the solition-antisoliton term ($s\bar{s}$) and the breather-breather term ($b_1 b_1$) for $\xi = 0.62734$ (left) and $\xi = 0.82734$ (right). The first column shows the exact values of $\Delta_{\mathcal{T}}$ and the last column the sum of $s\bar{s}$ and $b_1 b_1$ contributions. As expected, the main contribution comes from the $s\bar{s}$ term. This contribution gets larger as we approach the threshold value $\xi = 1$, while the breather contribution is reduced.

Let us consider the regimes when there are one, two, three or four breathers present we have that the expansion above can be approximated as follows:

- For $\xi > 1$ we are in the repulsive regime where no breathers are present. The main contribution to the $\Delta$ sum rule comes from the soliton-antisoliton form factor and was computed in [27].

- For $\frac{1}{2} \leqslant \xi < 1$ we have a single breather $b_1$ present and the main contributions are

$$\Delta_{\mathcal{T}}^{(1)} \approx -\frac{n}{32\pi^2 m^2 \langle \mathcal{T} \rangle} \int_{-\infty}^{\infty} d\theta \, \frac{4\sin^2\frac{\pi\xi}{2} \, F_{s\bar{s}}^{\mathcal{T}}(\theta; \xi, n) F_{s\bar{s}}^{\Theta}(\theta; \xi)^* + F_{b_1 b_1}^{\mathcal{T}}(\theta; \xi, n) F_{b_1 b_1}^{\Theta}(\theta; \xi)^*}{4\sin^2\frac{\pi\xi}{2}\cosh^2\frac{\theta}{2}} . \quad (73)$$

  The sum for two values of $\xi$ is presented in Table 1.

- For $\frac{1}{3} \leqslant \xi < \frac{1}{2}$ we have two breathers $b_1, b_2$ present and the main contributions are

$$\Delta_{\mathcal{T}}^{(2)} \approx \Delta_{\mathcal{T}}^{(1)} - \frac{n \, F_{b_2}^{\mathcal{T}}(\xi, n) F_{b_2}^{\Theta}(\xi)^*}{8\pi m^2 \sin^2 \pi\xi \langle \mathcal{T} \rangle} - \frac{n}{32\pi^2 m^2 \langle \mathcal{T} \rangle} \int_{-\infty}^{\infty} d\theta \, \frac{F_{b_2 b_2}^{\mathcal{T}}(\theta; \xi, n) F_{b_2 b_2}^{\Theta}(\theta; \xi)^*}{4\sin^2 \pi\xi \cosh^2\frac{\theta}{2}} . \quad (74)$$

  Numerical values of the sum (74) and of individual contributions to it are presented in Table 2 of Appendix E.

- For $\frac{1}{4} \leqslant \xi < \frac{1}{3}$ we have three breathers $b_1, b_2, b_3$ present and the main contributions are

$$
\begin{aligned}
\Delta_{\mathcal{T}}^{(3)} \;\approx\; \Delta_{\mathcal{T}}^{(2)} &- \frac{n}{32\pi^2 m^2 \langle \mathcal{T} \rangle} \int_{-\infty}^{\infty} d\theta \, \frac{F_{b_3 b_3}^{\mathcal{T}}(\theta; \xi, n) F_{b_3 b_3}^{\Theta}(\theta; \xi)^*}{4 \sin^2 \frac{3\pi\xi}{2} \cosh^2 \frac{\theta}{2}} \\
&- \frac{n}{64\pi^2 m^2 \langle \mathcal{T} \rangle} \int_{-\infty}^{\infty}\int_{-\infty}^{\infty} d\theta_1 d\theta_2 \, \frac{F_{b_1 b_3}^{\mathcal{T}}(\theta_1 - \theta_2; \xi, n) F_{b_1 b_3}^{\Theta}(\theta_1 - \theta_2; \xi)^*}{(\sin \frac{\pi\xi}{2} \cosh \theta_1 + \sin \frac{3\pi\xi}{2} \cosh \theta_2)^2}
\end{aligned}
\tag{75}
$$

- Finally, for $\frac{1}{5} \leqslant \xi < \frac{1}{4}$ we have four breathers $b_1, b_2, b_3, b_4$ present and the main contributions are

$$
\begin{aligned}
\Delta_{\mathcal{T}}^{(4)} \;\approx\; \Delta_{\mathcal{T}}^{(3)} &- \frac{n \, F_{b_4}^{\mathcal{T}}(\xi, n) F_{b_4}^{\Theta}(\xi)^*}{8\pi m^2 \sin^2 2\pi\xi \langle \mathcal{T} \rangle} - \frac{n}{32\pi^2 m^2 \langle \mathcal{T} \rangle} \int_{-\infty}^{\infty} d\theta \, \frac{F_{b_4 b_4}^{\mathcal{T}}(\theta; \xi, n) F_{b_4 b_4}^{\Theta}(\theta; \xi)^*}{4 \sin^2 2\pi\xi \cosh^2 \frac{\theta}{2}} \\
&- \frac{n}{64\pi^2 m^2 \langle \mathcal{T} \rangle} \int_{-\infty}^{\infty}\int_{-\infty}^{\infty} d\theta_1 d\theta_2 \, \frac{F_{b_2 b_4}^{\mathcal{T}}(\theta_1 - \theta_2; \xi, n) F_{b_2 b_4}^{\Theta}(\theta_1 - \theta_2; \xi)^*}{(\sin \pi\xi \cosh \theta_1 + \sin 2\pi\xi \cosh \theta_2)^2} \, .
\end{aligned}
\tag{76}
$$

Table 3 gives an example of the evaluation of the sum (76), albeit without including the $b_2 b_4$, $b_3 b_3$ and $b_4 b_4$ contributions, which we have not evaluated in this paper. Even so, the sum rule is approximately 95% saturated.

In conclusion, our numerical evaluation of the $\Delta$ sum rule in various regions of the attractive regime shows near saturation upon inclusion of all relevant one-particle and two-particle form factors and therefore provides strong backing for our analytical results. It is interesting to note that the deeper we go into the attractive regime (i.e. the smaller $\xi$ is) the more significant breather contributions are, so that for instance, in Table 2(d) the soliton-antisoliton contribution represents only about 20% of the total value of the dimension.

# 7    Application: Entanglement Oscillations after a Mass Quench

An interesting application of our results is to the study of the entanglement dynamics of the sine-Gordon model after a global mass quench [41, 42]. That is, we want to study the time-dependence of a certain measure of entanglement when the mass scale $m$ is abruptly changed at time zero. Then, if the original hamiltonian of the system was $H(m)$ and $m$ was the pre-quench soliton mass, at times $t > 0$ the system will time-evolve with a new Hamiltonian $H(\hat{m})$, where $\hat{m}$ is the post-quench soliton mass. In such a situation, the reduced density matrix may be formally written as:

$$
\rho_A = \mathrm{Tr}_B (e^{-itH(\hat{m})} |0\rangle\langle 0| e^{itH(\hat{m})}) \, ,
\tag{77}
$$

where $A$ and $B$ are two complementary regions and $|0\rangle$ is the pre-quench ground state. In terms of $\rho_A$ the Rényi and von Neumann entropies are defined in the usual form:

$$
S_n(t) := \frac{\log(\mathrm{Tr}\rho_A^n)}{1 - n} \, , \qquad S_1(t) := \lim_{n \to 1} S_n(t) \, ,
\tag{78}
$$

and if $A$ is a semi-infinite region, these expressions are equivalent to:

$$
S_n(t) = \frac{\log \left( \varepsilon^{2\Delta_{\mathcal{T}} n} {}_n\langle 0 | \mathcal{T}(0, t) | 0 \rangle_n \right)}{1 - n} \, ,
\tag{79}
$$

and its $n \to 1$ limit, where $\varepsilon$ is a non-universal UV cut-off which can be eliminated by considering instead the quantities

$$\Delta S_n(t) := S_n(t) - S_n(0). \qquad (80)$$

and $|0\rangle_n$ is the pre-quench ground state in the replica theory. Note that $S_n(0)$ is a function of the vacuum expectation value $_n\langle 0|\mathcal{T}(0,0)|0\rangle_n$ which we have abbreviated as $\langle \mathcal{T}\rangle$ in our form factor formulae.

With these definitions, the situation we want to consider here is entirely analogous to the studies performed in [43, 44]. In fact, the present model has two key common features with the minimal $E_8$ Toda field theory studied in [44]. They are the presence of non-vanishing one-particle form factors and a mass spectrum where all masses are proportional to a fundamental scale $m$ (the mass of the soliton/antisoliton). Carrying out the quench perturbation theory proposed in [45], non-vanishing one-particle form factors inevitably lead to entanglement oscillations at first order in perturbation theory. As observed in [43, 46] the dynamics of entanglement is closely tied to the dynamics of the one-point function of the order parameter. Indeed, oscillations of the one-point function of the order parameter in the sine-Gordon model, following a mass quench where found in [47] employing perturbation theory.

The formulae involved are almost identical to those presented in [44], specially in the supplementary material. We must just highlight that the field associated with the mass quench in this case is the perturbing field in the sine-Gordon theory, namely the field $\Psi = 2\cos g\varphi$ where $g$ is the coupling we first encountered in the action (1). This field is, as usual, proportional to the trace of the stress-energy tensor, hence its form factors are identical to those of $\Theta$ up to a proportionality constant (essentially, we need to replace $\langle \Theta\rangle = 2\pi m_1^2$ with $\langle \Psi\rangle$). Let us consider a perturbation where the original coupling $\mu$ in the action (1) is changed by a small amount $\delta_\mu$, that is $\mu \mapsto \mu + \delta_\mu$ with $\frac{\delta_\mu}{\mu} \ll 1$. Then, the $\mathcal{O}(\delta_\mu)$ contribution to the Rényi entropies may be expressed as a series in form factors of $\mathcal{T}$ and $\Psi$, where the leading contributions to the increment of the Rényi entropies, come from one- and two-particle form factors. After various simplifications, the series takes the form

$$\Delta S_n(t) = \frac{1}{1-n}\frac{\delta_\mu}{\mu}\left[\frac{2\Delta_{\mathcal{T}}}{2-2\Delta_\Psi} + n\,\mathcal{C}_\Psi \sum_{k=1}^{\left[\frac{\ell(\xi)}{2}\right]}\frac{2}{r_{2k}^2}\hat{F}_{b_{2k}}^\Psi(\xi)^*\hat{F}_{b_{2k}}^{\mathcal{T}}(\xi,n)\cos(r_{2k}\hat{m}t) \qquad (81)\right.$$

$$+2n\,\mathcal{C}_\Psi\int_{-\infty}^\infty\frac{\mathrm{d}\theta}{2\pi}\frac{\mathrm{Re}\left[[\hat{F}_{s\bar{s}}^\Psi(2\theta;\xi)]^*\hat{F}_{s\bar{s}}^{\mathcal{T}}(2\theta;\xi,n)e^{-2i\hat{m}t\cosh\theta}\right]}{2\cosh^2\theta}$$

$$+2n\,\mathcal{C}_\Psi\int_{-\infty}^\infty\frac{\mathrm{d}\theta}{2\pi}\sum_{k=1}^{\ell(\xi)}\frac{\mathrm{Re}\left[[\hat{F}_{b_kb_k}^\Psi(2\theta;\xi)]^*\hat{F}_{b_kb_k}^{\mathcal{T}}(2\theta;\xi,n)e^{-2ir_k\hat{m}t\cosh\theta}\right]}{2r_k^2\cosh^2\theta}$$

$$+2n\,\mathcal{C}_\Psi\int_{-\infty}^\infty\frac{\mathrm{d}\theta}{2\pi}\sum_{k\neq p}^{\prime}\frac{1}{r_k\cosh\theta(r_k\cosh\theta + r_p\cosh\tilde{\theta})}$$

$$\left.\times\mathrm{Re}\left[[\hat{F}_{b_kb_p}^\Psi(\theta-\tilde{\theta})]^*\hat{F}_{b_kb_p}^{\mathcal{T}}(\theta-\tilde{\theta})e^{-i\hat{m}t(r_k\cosh\theta+r_p\cosh\tilde{\theta})}\right] + \dots\right] + \mathcal{O}(\delta_\lambda^2),$$

where

$$\tilde{\theta} := -\sinh^{-1}\left(\frac{r_k}{r_p}\sinh\theta\right) , \tag{82}$$

$r_k = \frac{\hat{m}_k}{\hat{m}}$ are the scaled post-quench breather masses. The "prime" symbol in the last sum indicates the additional restriction that only terms where $k$ and $p$ are either both even or both odd will be non-vanishing. The "hatted" form factors are scaled versions of the usual form factors where the expectation values of the associated fields have been factored out. This dependency can then be absolved into the ratio of couplings $\delta_\mu/\mu$ and the constant $\mathcal{C}_\Psi$. The conformal dimension $\Delta_\Psi = g^2 = \frac{\xi}{1+\xi}$ and the constant

$$\mathcal{C}_\Psi = \frac{\mathcal{A}_\Psi}{\kappa^2} \qquad \text{where} \qquad \langle\Psi\rangle = \mathcal{A}_\Psi \mu^{\frac{2\Delta_\Psi}{2-2\Delta_\Psi}} \qquad \text{and} \qquad m = \kappa\mu^{\frac{1}{2-2\Delta_\Psi}} . \tag{83}$$

These are the standard scaling laws for vacuum expectation values and the mass-coupling relation. A relationship between the constant $\mathcal{A}_\Psi$ and $\kappa$ can be read off from the paper [48] where the expectation values of exponential fields in the sine-Gordon model were obtained. From this formula it follows that

$$\langle\Psi\rangle = 2\left[\frac{m\sqrt{\pi}\Gamma\left(\frac{1}{2-2g^2}\right)}{2\Gamma\left(\frac{g^2}{2-2g^2}\right)}\right]^{2g^2} \exp\int_0^\infty \frac{dt}{t}\left[\frac{\sinh^2(2g^2t)}{2\sinh(g^2t)\sinh t\cosh((1-g^2)t)} - 2g^2e^{-2t}\right] . \tag{84}$$

It is important to note that this formula is only convergent for $g^2 < \frac{1}{2}$, which excludes the repulsive regime [48]. The mass-coupling relation was given earlier in (4). This allows us to fix the ratio above to

$$\mathcal{C}_\Psi = \frac{\Gamma(g^2)\Gamma(\frac{1}{2-2g^2})^2}{2\Gamma(1-g^2)\Gamma(\frac{g^2}{2-2g^2})^2} \exp\int_0^\infty \frac{dt}{t}\left[\frac{\sinh^2(2g^2t)}{2\sinh(g^2t)\sinh t\cosh((1-g^2)t)} - 2g^2e^{-2t}\right] . \tag{85}$$

Despite the messy nature of the formula (81) (a very similar formula can be written for the von Neumann entropy) the main features of entanglement are rather clear: for small quenches, there will be undamped oscillations whenever any one-particle form factors are non-vanishing, confirming the general ideas observed in [44, 45]. In addition, there will be additional oscillatory terms coming from higher particle form factors which will be suppressed by a power of $t$ that depends on the leading behaviour of the form factors near zero rapidity (this can be analysed further by using a saddle-point approximation). This means that the dynamics of entanglement following a mass quench is rather different in the regime $\frac{1}{2} \geqslant \xi$ (undamped oscillations with at least two breathers present) and for for $\xi < \frac{1}{2}$ (damped oscillations with at most one breather present).

We demonstrate these qualitative differences in the entanglement evolution by evaluating (81) numerically for various values of $n$ and two particular values of $\xi$. In Figure 2 $\Delta S_n(t)$ is displayed for $n = 2, 3, 4, 5$ and for $\xi = 0.810361$ and $\xi = 0.420712$. Clearly, above the second breather threshold ($\xi = 0.810361$) no undamped oscillations can be seen, which are, very clearly present when the second breather joins the spectrum ($\xi = 0.420712$). Note that although $\Delta S_n(0) = 0$ by the definition (80), it is not exactly zero numerically (although it is rather small).

This is because $S_n(t)$ is evaluated at first order in perturbation theory and therefore its value at zero is only an approximation of the exact analytic value $S_n(0)$ that is subtracted in (80).

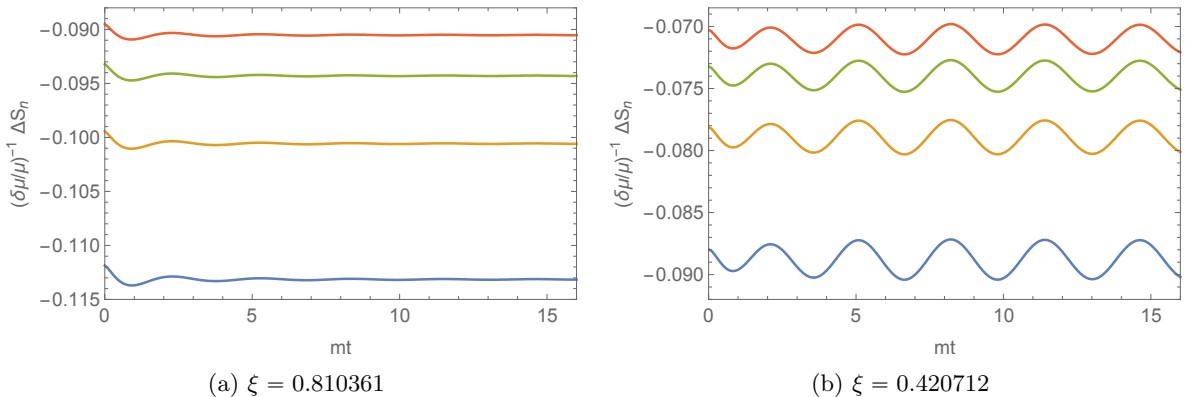

(a) $\xi = 0.810361$          (b) $\xi = 0.420712$

Figure 2: The time evolution of the rescaled Rényi entropy difference $\left(\frac{\delta\mu}{\mu}\right)^{-1}\Delta S_n$ after a mass quench in the sine-Gordon model with interaction parameter $\xi = 0.810361$ (a) and $\xi = 0.420712$ (b). Time is measured in units of the inverse soliton mass $m$ and the blue, yellow, green, and red curves correspond to Rényi entropies with $n = 2, 3, 4$ and $5$, respectively.

Note that these results are only expected to hold for small quenches and times $t < \mu^{-1}$, as explained in [45]. In addition, we know that for large times the leading feature of entanglement (in any regime) is linear growth [41, 42, 49]. As observed in other studies, this feature is not recovered using first-order perturbation theory as it is a second order effect [43, 44]. In addition, in some cases, like for $E_8$ Toda field theory, linear growth is very slow so that it only becomes apparent for large times in numerical simulations [44]. The same phenomenology is also observed for the von Neumann entropy for the same reasons.

It is worth considering whether or not these behaviours will persist for larger times and quenches. In this regard, arguments have been put forward as to why the undamped oscillations found at first order should be damped when including higher order terms [16]. At the same time, we know of at least one theory, $E_8$ Toda field theory, where this eventual damping is not observed numerically even for large quenches and times [44]. This suggests that this phenomenology still needs to be better understood. Similar behaviours have been observed in [16, 17, 47] for the expectation value of the field $\Psi$ and in [15] for two-point functions of the field $\varphi$.

# 8   Conclusion

In this paper we have carried out an in-depth study of the form factors of the branch point twist field in the sine-Gordon model in the attractive regime. We have considered up to four breathers in the spectrum and focused on the one- and two-particle form factors. Our work extends results for the repulsive regime that were presented in [27].

Although computations are generally tedious, great simplification comes from the presence of $U(1)$ symmetry in the soliton-antisoliton sector and $\mathbb{Z}_2$ symmetry in the breather sector. The

latter imply the vanishing of a large number of form factors so that only form factors containing the same number of solitons and antisolitons as well as an even number of odd breathers are non-vanishing. The two-particle form factors of the soliton-antisoliton sector can be computed by diagonalizing the form factor equations (as the theory is non-diagonal) and incorporating the correct structure of bound state and kinematic poles, as discussed in great detail in Section 4 and Appendix D. For the breather sector the combination of the fusion mechanism with the analytic continuation from sinh-Gordon, provide an effective way of constructing the form factors of heavier breathers from those of lighter ones leading to the results of Secion 5 and Appendices B and C.

Our form factors can now be employed to compute correlation functions of branch point twist fields, hence a number of entanglement measures. In this paper we have highlighted just one such application, namely to the study of the entanglement dynamics following a mass quench in the sine-Gordon model. As observed in a similar study [44] we find that at least for small quenches, undamped oscillations of frequencies proportional to the even breather masses are present and constitute the leading behaviour of Rényi and von Neumann entropies. This is analogous to results found in [16,17,47] for the expectation value of the field $\Psi$ and more generally in [15] for two-point functions of the field $\varphi$.

As anticipated in the introduction, our immediate goal now is to extend these results to symmetry resolved twist fields [25, 26].

**Acknowledgment:** We are grateful to Benjamin Doyon, Gábor Takács and Jacopo Viti for useful discussions. We specially thank Pasquale Calabrese for discussions and for his early stage involvement in this project. DXH acknowledges support from ERC under Consolidator grant number 771536 (NEMO).

# A   Minimal Form Factors Mixed Representations

As mentioned in Sections 4 and 5 the most useful representation for the minimal form factors is neither exponential nor based entirely on Gamma-function products, but a mixture of the two. This idea was employed first in [22] and can be implemented in a similar way for any minimal form factor of the type described in this paper. The function (36) can be written as

$$
\Phi(\theta;\xi,n) = -i \sinh\frac{\theta}{2n} \prod_{k,p=0}^{N} \left[ \frac{\Gamma\left(\frac{p+n+(k+1)\xi}{2n}\right)^2 \Gamma\left(\frac{n+\frac{i\theta}{\pi}+p+1+k\xi+n}{2n}\right) \Gamma\left(\frac{-n-\frac{i\theta}{\pi}+p+1+k\xi+n}{2n}\right)}{\Gamma\left(\frac{p+n+k\xi+1}{2n}\right)^2 \Gamma\left(\frac{n+\frac{i\theta}{\pi}+p+(k+1)\xi+n}{2n}\right) \Gamma\left(\frac{-n-\frac{i\theta}{\pi}+p+(k+1)\xi+n}{2n}\right)} \right]^{(-1)^p}
$$
$$
\times \exp\left[ -\int_0^\infty \frac{dt}{t} \frac{\sinh\left(\frac{1}{2}(1-\xi)t\right)\left(e^{-\xi(N+1)t} + e^{-(\xi+1)(N+1)t} - e^{-(N+1)t}\right)\sinh^2\left(\frac{t}{2}\left(n-\frac{\theta}{i\pi}\right)\right)}{\cosh\frac{t}{2}\sinh\frac{\xi t}{2}\sinh nt} \right].
$$
$$
(86)
$$

Whereas the functions $\varphi_{\pm}(\theta; \xi, n)$ can be written as

$$\varphi_+(\theta; \xi, n) = \prod_{p=0}^{N} \frac{\Gamma\left(\frac{n+2p\xi+1}{2n}\right)^2 \Gamma\left(\frac{-\frac{i\theta}{\pi}+2p\xi+2\xi-1}{2n}\right) \Gamma\left(\frac{2n+\frac{i\theta}{\pi}+2p\xi+2\xi-1}{2n}\right)}{\Gamma\left(\frac{n+2p\xi+2\xi-1}{2n}\right)^2 \Gamma\left(\frac{-\frac{i\theta}{\pi}+2p\xi+1}{2n}\right) \Gamma\left(\frac{2n+\frac{i\theta}{\pi}+2p\xi+1}{2n}\right)}$$
$$\times \exp\left[-2\int_0^{\infty} \frac{dt}{t} \frac{e^{-2\xi(N+1)t} \sinh((\xi-1)t) \sinh^2\left(\frac{t}{2}\left(n-\frac{\theta}{i\pi}\right)\right)}{\sinh(nt)\sinh(\xi t)}\right] \tag{87}$$

This quantity is independent of the choice of $N$ only when $\xi > 1/2$, but can be made convergent for any positive real values of $\xi$ if the minimal allowed value for $N$ is suitably chosen. In fact (87) gives a physically motivated (as seen in Appendix D) and correct analytic continuation. Alternatively, for $\frac{1}{2p} \geqslant \xi > \frac{1}{2p+2}$ and $p \in \mathbb{Z}^+$

$$\varphi_-(\theta; \xi, n) = \prod_{m=0}^{N} \frac{\Gamma\left(\frac{n+2(m-p)\xi+1}{2n}\right)^2 \Gamma\left(\frac{-\frac{i\theta}{\pi}+2(m+p+1)\xi-1}{2n}\right) \Gamma\left(\frac{2n+\frac{i\theta}{\pi}+2(m+p+1)\xi-1}{2n}\right)}{\Gamma\left(\frac{n+2(m+p+1)\xi-1}{2n}\right)^2 \Gamma\left(\frac{-\frac{i\theta}{\pi}+2(m-p)\xi+1}{2n}\right) \Gamma\left(\frac{2n+\frac{i\theta}{\pi}+2(m-p)\xi+1}{2n}\right)}$$
$$\times \exp\left[-2\int_0^{\infty} \frac{dt}{t} \frac{e^{-2\xi(N+1)t} \sinh(((2p+1)\xi-1)t) \sinh^2\left(\frac{t}{2}\left(n-\frac{\theta}{i\pi}\right)\right)}{\sinh(nt)\sinh(\xi t)}\right]. \tag{88}$$

Finally, the breather-breather minimal form factor also admits the representation

$$R(\theta; \xi, n) = \prod_{k=0}^{N} \left[\frac{\Gamma\left(\frac{-\frac{i\theta}{\pi}-\xi+k}{2n}\right) \Gamma\left(1+\frac{\frac{i\theta}{\pi}-\xi+k}{2n}\right) \Gamma\left(\frac{-\frac{i\theta}{\pi}+1+\xi+k}{2n}\right) \Gamma\left(1+\frac{\frac{i\theta}{\pi}+1+\xi+k}{2n}\right)}{\Gamma\left(\frac{-\frac{i\theta}{\pi}+k}{2n}\right) \Gamma\left(1+\frac{\frac{i\theta}{\pi}+k}{2n}\right) \Gamma\left(\frac{-\frac{i\theta}{\pi}+k+1}{2n}\right) \Gamma\left(1+\frac{\frac{i\theta}{\pi}+k+1}{2n}\right)}\right]^{(-1)^k}$$
$$\times \exp\left[4\int_0^{\infty} \frac{dt}{t} \frac{e^{-\frac{t}{2}(3+4N)} \sinh\frac{\xi t}{2} \sinh\frac{(1+\xi)t}{2} \cosh t\left(n+\frac{i\theta}{\pi}\right)}{(1+e^t)\sinh(nt)}\right]. \tag{89}$$

# B  Computation of Branch Point Twist Field Breather Form Factors from Fusion

## B.1  Computation of $F_{b_3 b_1}(\theta; \xi, n)$

Using fusion again we have that

$$-i\operatorname*{Res}_{\theta=\theta_1} F_{b_2 b_1 b_1}\left(\theta + \frac{i\pi\xi}{2}, \theta_1 - i\pi\xi, \theta_2; \xi, n\right) = \Gamma_{b_2 b_1}^{b_3} F_{b_3 b_1}(\theta_{12}; \xi, n). \tag{90}$$

This gives a solution of the form

$$F_{b_3 b_1}(\theta_{12}; \xi, n) = H_{31}(\xi, n) Q_{31}(x_1, x_2; \xi, n) \frac{R(\theta_{12}; \xi, n) R(\theta_{12}+i\pi\xi; \xi, n) R(\theta_{12}-i\pi\xi; \xi, n)}{(x_1 - x_2\omega\beta)(x_2 - x_1\omega\beta)(x_1\omega - x_2)(x_2\omega - x_1\beta)}. \tag{91}$$

The polynomial $Q_{31}(x_1, x_2; \xi, n)$ follows from the reduction of $Q_{211}(x_1\beta^{-\frac{1}{2}}, x_1\beta, x_2; \xi, n)$ and can be written as

$$Q_{31}(x_1, x_2; \xi, n) = x_1 x_2 (\omega x_1 - x_2)(x_2 \omega - x_1 \beta) , \tag{92}$$

if we also identify

$$H_{31}(\xi, n) = -\langle \mathcal{T} \rangle \frac{2\omega\beta \sin \frac{\pi}{2n} \sin \frac{\pi}{n} \cos \frac{\pi(1-\xi)}{2n}(1 + 2\cos \frac{\pi\xi}{n})\Gamma^{b_3}_{b_2 b_1}\Gamma^{b_2}_{b_1 b_1}}{n^2 \sin \frac{\pi(1+\xi)}{2n} \sin \frac{\pi(1-2\xi)}{2n}} \frac{R(-2\pi i\xi; \xi, n)R(-i\pi\xi; \xi, n)^2}{R(i\pi; \xi, n)^2} . \tag{93}$$

## B.2 Computation of $F_{b_4}(\xi, n)$ from fusion in $F_{b_3 b_1}(\theta; \xi, n)$

Computing

$$-i \operatorname*{Res}_{\theta = 2\pi i\xi} F_{b_3 b_1}(\theta; \xi, n) = \Gamma^{b_4}_{b_3 b_1} F_{b_4}(\xi, n) , \tag{94}$$

which gives

$$
\begin{aligned}
F_{b_4}(\xi, n) &= \langle \mathcal{T} \rangle \frac{\sin \frac{\pi}{n} \sin \frac{\pi}{2n}(1 + 2\cos \frac{\pi\xi}{n}) \cos \frac{\pi(1-\xi)}{2n} \Gamma^{b_4}_{b_3 b_1}\Gamma^{b_3}_{b_2 b_1}\Gamma^{b_2}_{b_1 b_1}}{2n^2 \sin^2 \frac{\pi(1+\xi)}{2n} \sin \frac{\pi(1-2\xi)}{2n} \sin \frac{\pi(1-3\xi)}{2n}} \\
&\times \frac{R(-3\pi i\xi; \xi, n)R(-2\pi i\xi; \xi, n)^2 R(-i\pi\xi; \xi, n)^3}{R(i\pi; \xi, n)^2} ,
\end{aligned}
\tag{95}
$$

which is plotted in Fig. 3 as a function of $\xi$ and $n$.

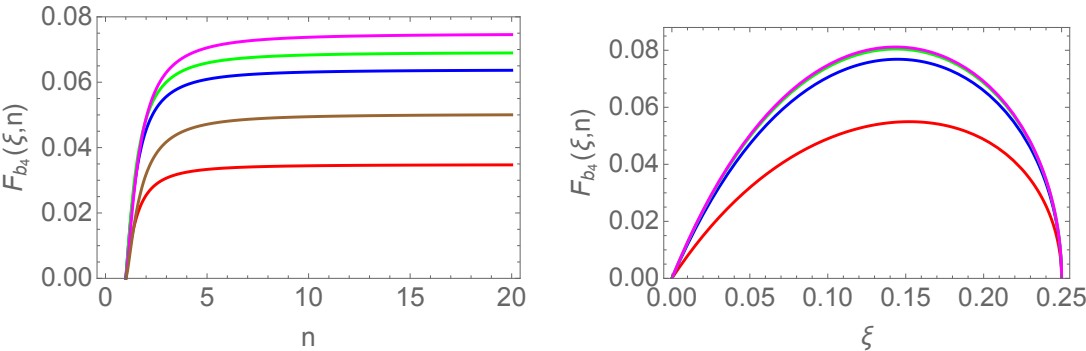

Figure 3: Left: The one-particle form factor $F_{b_4}(\xi, n)$ as a function of $n$ for $\xi = 0.24$ (red), 0.21 (blue), 0.2 (green), 0.1 (magenta) and 0.05 (brown). Right: The one-particle form factor $F_{b_4}(\xi, n)$ as a function of $\xi$ for $n = 2$ (red), 5 (blue), 12 (green), 50 (magenta).

We have also that

$$
\begin{aligned}
\lim_{n \to 1} \frac{F_{b_4}(\xi, n)}{1 - n} &= \frac{2\pi \sin^4 \frac{\pi\xi}{2} \Gamma^{b_4}_{b_3 b_1}\Gamma^{b_3}_{b_2 b_1}\Gamma^{b_2}_{b_1 b_1}}{\sin 2\pi\xi \sin^2 \pi\xi} \frac{1 + 2\cos \pi\xi}{1 - 2\cos \pi\xi} \\
&\times \frac{R(-3\pi i\xi; \xi, 1)R(-2\pi i\xi; \xi, 1)^2 R(-i\pi\xi; \xi, 1)^3}{R(i\pi; \xi, 1)^2} .
\end{aligned}
\tag{96}
$$

Note that the breather $b_4$ is only present for $\xi < \frac{1}{4}$.

## B.3   Computation of $F_{b_2b_2}(\theta;\xi,n)$

Starting with the form factor $F_{b_2b_1b_1}(\theta_1,\theta_2,\theta_2;\xi,n)$ we can now fuse the last two particles to obtain $F_{b_2b_2}(\theta;\xi,n)$. The bound state residue equation dictates that

$$-i\operatorname*{Res}_{\theta=\theta_1}F_{b_1b_1b_2}(\theta+\frac{i\pi\xi}{2},\theta_1-\frac{i\pi\xi}{2},\theta_2;\xi,n)=\Gamma^{b_2}_{b_1b_1}F_{b_2b_2}(\theta_1,\theta_2;\xi,n)\,, \tag{97}$$

From the the form factor axioms we can write the following

$$-i\operatorname*{Res}_{\theta_0=\theta_1}F_{b_1b_1b_2}(\theta_0+\frac{i\pi\xi}{2},\theta_1-\frac{i\pi\xi}{2},\theta_2;\xi)=\Gamma^{b_2}_{b_1b_1}F_{b_2b_2}(\theta_{12};\xi) \tag{98}$$

$$=-i\operatorname*{Res}_{\theta_0=\theta_1}F_{b_2b_1b_1}(\theta_2,\theta_1-\frac{i\pi\xi}{2},\theta_0+\frac{i\pi\xi}{2};\xi)S_{b_1b_2}(\theta_{12}-\frac{i\pi\xi}{2})S_{b_1b_2}(\theta_{02}+\frac{i\pi\xi}{2})S_{b_1b_1}(\theta_{01}+i\pi\xi)$$

$$=(\Gamma^{b_2}_{b_1b_2})^2F_{b_2b_1b_1}(\theta_2,\theta_1-\frac{i\pi\xi}{2},\theta_1+\frac{i\pi\xi}{2};\xi)S_{b_1b_2}(\theta_{12}-\frac{i\pi\xi}{2})S_{b_1b_2}(\theta_{12}+\frac{i\pi\xi}{2})$$

$$=(\Gamma^{b_2}_{b_1b_2})^2S_{b_2b_2}(\theta_{12})F_{b_2b_1b_1}(\theta_2,\theta_1-\frac{i\pi\xi}{2},\theta_1+\frac{i\pi\xi}{2};\xi)\,. \tag{99}$$

where we used the bootstrap equation for the breather $S$-matrices

$$S_{b_1b_2}(\theta-\frac{i\pi\xi}{2})S_{b_1b_2}(\theta+\frac{i\pi\xi}{2})=S_{b_2b_2}(\theta)\,. \tag{100}$$

Then it immediately follows that the two-particle form factor has the following structure

$$F_{b_2b_2}(\theta_{12};\xi,n)=H_{22}(\xi,n)Q_{22}(x_1,x_2;\xi,n)\frac{R(\theta_{12};\xi,n)^2R(\theta_{12}+i\pi\xi;\xi,n)R(\theta_{12}-i\pi\xi;\xi,n)}{(x_1-\omega x_2)(x_2-\omega x_1)(x_1-\omega\beta x_2)(x_2-\omega\beta x_1)}\,, \tag{101}$$

with

$$Q_{22}(x_1,x_2;\xi,n)=\alpha_1(\xi,n)\sigma_1^4+\alpha_2(\xi,n)\sigma_2\sigma_1^2+\alpha_3(\xi,n)\sigma_2^2\,, \tag{102}$$

$$\begin{aligned}\alpha_1(\xi,n)&=\omega\beta^2(1+\omega)\,,\\ \alpha_2(\xi,n)&=-(\beta(1+\beta)+\omega\beta^2(\beta+\beta^2+4)+\omega^2(4\beta^2+\beta+1)+\omega^3\beta^2)\,,\\ \omega_3(\xi,n)&=-1+\omega^2(\beta^5+5\beta^2+2\beta^4+3\beta^3+2\beta+1)+(\omega^{-1}+\omega^4)\beta(1+\beta+\beta^2)\\ &\quad+\omega(3\beta+2\beta^3+\beta^4+5\beta^2+2+\beta^{-1})-\omega^3\beta^4\,,\end{aligned} \tag{103}$$

and

$$H_{22}(\xi,n)=\langle\mathcal{T}\rangle\frac{\sqrt{\omega}\beta^{-1}\sin\frac{\pi}{n}\sin\frac{\pi}{2n}(\Gamma^{b_2}_{b_1b_1})^2}{4n^2\sin^2\frac{\pi(\xi-1)}{2n}\sin^2\frac{\pi(\xi+1)}{2n}}\frac{R(-i\pi\xi;\xi,n)^2}{R(i\pi;\xi,n)^2}=\frac{\sqrt{\omega}F_{b_2}(\xi,n)^2}{2\beta\cos\frac{\pi}{2n}\langle\mathcal{T}\rangle}\,. \tag{104}$$

## B.4   Computation of $F_{b_4}$ from Fusion in $F_{b_2b_2}(\theta;\xi,n)$

Finally, we may consider the fusion of two $b_2$ breathers to form $b_4$. We employ the equation

$$-i\operatorname*{Res}_{\theta=\theta_1}F_{b_2b_2}(\theta+i\pi\xi,\theta_1-i\pi\xi;\xi,n)=\Gamma^{b_4}_{b_2b_2}F_{b_4}(\xi,n)\,, \tag{105}$$

This gives us

$$
\begin{aligned}
F_{b_4}(\xi, n) \; = \; \langle \mathcal{T} \rangle & \frac{\sin \frac{\pi}{n} \sin \frac{\pi}{2n} (1 + 2 \cos \frac{\pi \xi}{n}) \cos \frac{\pi(1-\xi)}{2n} (\Gamma^{b_2}_{b_1 b_1})^2 \Gamma^{b_4}_{b_2 b_2}}{2n^2 \sin \frac{(1-2\xi)\pi}{2n} \sin \frac{(1-3\xi)\pi}{2n} \sin^2 \frac{(1+\xi)\pi}{2n}} \\
& \times \frac{R(-3\pi i \xi; \xi, n) R(-2\pi i \xi; \xi, n)^2 R(-i\pi \xi; \xi, n)^3}{R(i\pi; \xi, n)^2} \, .
\end{aligned}
\tag{106}
$$

This is identical to the result we obtained from fusing $F_{b_3 b_1}(\theta; \xi, n)$ with the identification

$$
\Gamma^{b_4}_{b_3 b_1} \Gamma^{b_3}_{b_2 b_1} \; = \; \Gamma^{b_4}_{b_2 b_2}
\tag{107}
$$

## C    Form Factors of the Stress-Energy Tensor from Fusion

The form factors of the trace of the stress-energy tensor in the sinh-Gordon model where first computed in [22], where closed formulae for special values of the coupling were obtained. We are interested in the case of generic coupling $B$ for which solutions up to 14 particles where given. These solutions will be the building blocks for our fusion procedure. We are particularly interested in the one-particle form factors of the second and fourth breather which requires the two- and four-particle form factors of the stress energy tensor in sinh-Gordon. Replacing $B = -2\xi$ these form factors are given by

$$
F^{\Theta}_{b_1 b_1}(\theta; \xi) = 2\pi m_1^2 \frac{R(\theta; \xi, 1)}{R(i\pi; \xi, 1)} \, ,
\tag{108}
$$

where $m_1$ is the mass of the first breather as given in (3) and

$$
F^{\Theta}_{b_1 b_1 b_1 b_1}(\theta_1, \theta_2, \theta_3, \theta_4; \xi) = -\frac{8\pi m_1^2 \sin \pi \xi}{R(i\pi; \xi, 1)^2} \sigma_1 \sigma_2 \sigma_3 \prod_{1 \leqslant i < j \leqslant 4} \frac{R(\theta_{ij}; \xi, 1)}{x_i + x_j} \, ,
\tag{109}
$$

with $x_i = e^{\theta_i}$ and $\sigma_i$ the elementary symmetric polynomial on variables $\{x_1, x_2, x_3, x_4\}$.

### C.1    Computation of $F^{\Theta}_{b_2}(\xi)$

Applying the fusion procedure to the two-particle form factor we have that

$$
-i \operatorname*{Res}_{\theta = i\pi \xi} F^{\Theta}_{b_1 b_1}(\theta; \xi) = \Gamma^{b_2}_{b_1 b_1} F^{\Theta}_{b_2}(\xi) \, ,
\tag{110}
$$

and we get simply

$$
F^{\Theta}_{b_2}(\xi) = 2\pi m_1^2 \sqrt{2 \tan \pi \xi} \frac{R(-i\pi \xi; \xi, 1)}{R(i\pi; \xi, 1)} \, ,
\tag{111}
$$

which is plotted in Fig. 4.

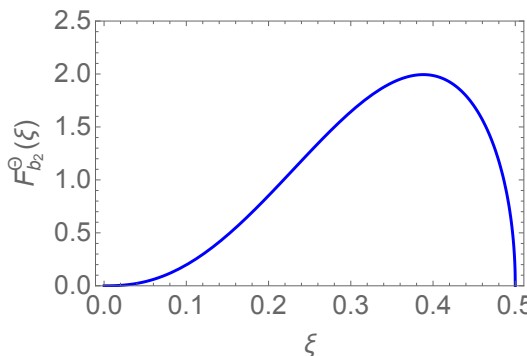

Figure 4: The one particle form factor $F_{b_2}^{\Theta}(\xi)$ for $m = 1$.

## C.2 Computation of $F_{b_2 b_1 b_1}^{\Theta}(\theta_1, \theta_2, \theta_3; \xi)$

In order to get higher breather form factors we must use the four-particle solution above. For instance we may fuse the first two particles to obtain $F_{b_2 b_1 b_1}^{\Theta}(\theta_1, \theta_2, \theta_3; \xi)$. The relevant equation is

$$-i \operatorname*{Res}_{\theta_0 = \theta_1} F_{b_1 b_1 b_1 b_1}^{\Theta}\left(\theta_0 + \frac{i\pi\xi}{2}, \theta_1 - \frac{i\pi\xi}{2}, \theta_2, \theta_3; \xi\right) = \Gamma_{b_1 b_1}^{b_2} F_{b_2 b_1 b_1}^{\Theta}(\theta_1, \theta_2, \theta_3; \xi). \tag{112}$$

which after some simplifications gives

$$F_{b_2 b_1 b_1}^{\Theta}(\theta_1, \theta_2, \theta_3; \xi) = H_{211}^{\Theta}(\xi) Q_{211}^{\Theta}(x_1, x_2, x_3; \xi) \frac{R(\theta_{23}; \xi, 1)}{x_2 + x_3}$$

$$\times \frac{R(\theta_{12} + \frac{i\pi\xi}{2}; \xi, 1) R(\theta_{12} - \frac{i\pi\xi}{2}; \xi, 1) R(\theta_{13} - \frac{i\pi\xi}{2}; \xi, 1) R(\theta_{13} + \frac{i\pi\xi}{2}; \xi, 1)}{(x_2 + \alpha x_1)(x_1 + \alpha x_2)(x_3 + \alpha x_1)(x_1 + \alpha x_3)}. \tag{113}$$

with $\alpha := e^{\frac{i\pi\xi}{2}}$ and

$$Q_{211}^{\Theta}(x_1, x_2, x_3; \xi) = \left(\sigma_1 + 2\cos\frac{\pi\xi}{2}\hat{\sigma}_1\right)\left(\sigma_2 + 2\cos\frac{\pi\xi}{2}\sigma_1\hat{\sigma}_1 + \hat{\sigma}_1^2\right)\left(\hat{\sigma}_1\sigma_1 + 2\cos\frac{\pi\xi}{2}\sigma_2\right), \tag{114}$$

for $\hat{\sigma}_1 = x_1$, $\sigma_1 = x_2 + x_3$ and $\sigma_2 = x_2 x_3$. The normalization constant is

$$H_{211}^{\Theta}(\xi) = -\frac{8\pi m_1^2 \alpha^2 \sin\frac{\pi\xi}{2} R(-i\pi\xi; \xi, 1)\Gamma_{b_1 b_1}^{b_2}}{R(i\pi; \xi, 1)^2}. \tag{115}$$

## C.3 Computation of $F_{b_2 b_2}^{\Theta}(\theta; \xi)$

We know from Watson's equation that

$$F_{b_1 b_1 b_2}^{\Theta}(\theta_1, \theta_2, \theta_3; \xi) = F_{b_2 b_1 b_1}^{\Theta}(\theta_3, \theta_2, \theta_1; \xi) S_{b_1 b_2}(\theta_{23}) S_{b_1 b_2}(\theta_{13}) S_{b_1 b_1}(\theta_{12}) \tag{116}$$

So we have that

$$-i \operatorname*{Res}_{\theta_0 = \theta_1} F_{b_1 b_1 b_2}^{\Theta}\left(\theta_0 + \frac{i\pi\xi}{2}, \theta_1 - \frac{i\pi\xi}{2}, \theta_2; \xi\right) = \Gamma_{b_1 b_1}^{b_2} F_{b_2 b_2}^{\Theta}(\theta_{12}; \xi) \tag{117}$$

$$= (\Gamma_{b_1 b_2}^{b_2})^2 S_{22}(\theta_{12}) F_{b_2 b_1 b_1}^{\Theta}\left(\theta_2, \theta_1 - \frac{i\pi\xi}{2}, \theta_1 + \frac{i\pi\xi}{2}; \xi\right). \tag{118}$$

which follows exactly as in (99). This gives

$$F_{b_2b_2}^{\Theta}(\theta_{12};\xi) = H_{22}^{\Theta}(\xi)Q_{22}^{\Theta}(x_1,x_2;\xi)\frac{R(\theta_{12};\xi,1)^2R(\theta_{12}+i\pi\xi;\xi,1)R(\theta_{12}-i\pi\xi;\xi,1)}{(x_1+\alpha^2x_2)(x_2+\alpha^2x_1)}\,,\qquad (119)$$

where

$$H_{22}^{\Theta}(\xi) = -\frac{8\pi m_1^2\alpha^2\sin\pi\xi R(-i\pi\xi;\xi,1)^2(\Gamma_{b_1b_1}^{b_2})^2}{R(i\pi;\xi,1)^2}\,,\qquad Q_{22}^{\Theta}(x_1,x_2;\xi) = \sigma_1^2+2\cos\pi\xi\,\sigma_2\,,\qquad (120)$$

with $\sigma_1 = x_1+x_2$ and $\sigma_2 = x_1x_2$.

## C.4   Computation of $F_{b_4}^{\Theta}(\xi)$

By computing the residue

$$-i\operatorname*{Res}_{\theta=2\pi i\xi}F_{b_2b_2}^{\Theta}(\theta;\xi) = \Gamma_{b_2b_2}^{b_4}F_{b_4}^{\Theta}(\xi)\,,\qquad (121)$$

which gives

$$\begin{aligned}F_{b_4}^{\Theta}(\xi) &= -4\pi m_1^2\sec\frac{3\pi\xi}{2}\left(\sin\frac{\pi\xi}{2}+\sin\frac{5\pi\xi}{2}\right)(\Gamma_{b_1b_1}^{b_2})^2\Gamma_{b_2b_2}^{b_4}\\ &\quad\times\frac{R(-3\pi i\xi;\xi,1)R(-2\pi i\xi;\xi,1)^2R(-\pi i\xi;\xi,1)^3}{R(i\pi;\xi,1)^2}\,.\end{aligned}\qquad (122)$$

A plot of $F_{b_4}^{\Theta}(\xi)$ as a function of $\xi$ is presented in Fig. 5.

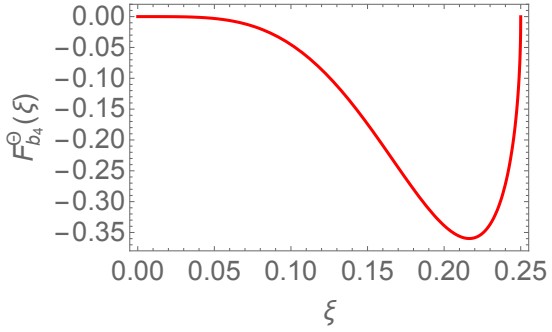

Figure 5: The one-particle form factor $F_{b_4}^{\Theta}(\xi)$ for $m=1$.

## C.5   Computation of $F_{b_3b_1}^{\Theta}(\theta;\xi)$

The last two-particle form factor that we can obtain starting with (109) is $F_{b_1b_3}^{\Theta}(\theta;\xi)$, resulting from the fusion process:

$$-i\operatorname*{Res}_{\theta_1=\theta_0}F_{b_2b_1b_1}^{\Theta}\left(\theta_1+\frac{i\pi\xi}{2},\theta_0-i\pi\xi,\theta_2;\xi\right)\qquad (123)$$

$$F_{b_3 b_1}^{\Theta}(\theta_{12}; \xi) = H_{31}^{\Theta}(\xi) Q_{31}^{\Theta}(x_1, x_2; \xi) \frac{R(\theta_{12}; \xi, 1) R(\theta_{12} - i\pi\xi; \xi, 1) R(\theta_{12} + i\pi\xi; \xi, 1)}{(x_1 + \alpha^2 x_2)(x_2 + \alpha^2 x_1)}, \tag{124}$$

with

$$H_{31}^{\Theta}(\xi) = -\frac{8\pi m_1^2 \alpha^2 \sin\frac{\pi\xi}{2} \sin\frac{3\pi\xi}{2}}{\sin 2\pi\xi} \frac{R(-i\pi\xi; \xi, 1)^2 R(-2\pi i\xi; \xi, 1) \Gamma_{b_1 b_1}^{b_2} \Gamma_{b_2 b_1}^{b_3}}{R(i\pi; \xi, 1)^2}, \tag{125}$$

$$Q_{31}^{\Theta}(x_1, x_2; \xi) = (x_1 + x_2 + 2x_2 \cos\pi\xi)(x_1 + x_2 + 2x_1 \cos\pi\xi). \tag{126}$$

# D   Dynamical Poles of the Soliton-Antisoliton Form Factors

In this appendix we first show that the two representations $G_{\pm}(\theta; \xi, n)$ of the minimal soliton-antisoliton form factor are indeed proportional to each other when the proper CDD-factors accounting for the bound state poles are introduced. We also demonstrate the precise working of the dynamical pole axiom (33).

Considering the first point, as $G_{\pm}(\theta; \xi, n) = \varphi_{\pm}(\theta; \xi, n)\Phi(\theta; \xi, n)$, it is enough to show that

$$\varphi_+(\theta; \xi, n) = \text{const} \times \prod_{k=1}^{[\frac{1}{2\xi}]} \frac{\cos\frac{\pi}{n} - \cos\frac{\pi(1-2k\xi)}{n}}{\cosh\frac{\theta}{n} - \cos\frac{\pi(1-2k\xi)}{n}} \varphi_-(\theta; \xi, n). \tag{127}$$

From Eqs. (37) and (38) we have that

$$
\begin{aligned}
\frac{\varphi_+(\theta; \xi, n)}{\varphi_-(\theta; \xi, n)} &= \prod_{k=1}^{[\frac{1}{2\xi}]} \frac{\Gamma\left(\frac{-\frac{i\theta}{\pi} - 2k\xi + 1}{2n}\right) \Gamma\left(1 + \frac{\frac{i\theta}{\pi} - 2k\xi + 1}{2n}\right) \Gamma\left(\frac{-\frac{i\theta}{\pi} + 2k\xi - 1}{2n}\right) \Gamma\left(1 + \frac{\frac{i\theta}{\pi} + 2k\xi - 1}{2n}\right)}{\Gamma\left(\frac{n - 2k\xi + 1}{2n}\right)^2 \Gamma\left(\frac{n + 2k\xi - 1}{2n}\right)^2}, \\
&= \prod_{k=1}^{[\frac{1}{2\xi}]} \frac{\cos\left(\frac{\pi(1-2k\xi)}{n}\right) + 1}{\cos\left(\frac{\pi(1-2k\xi)}{n}\right) - \cosh\frac{\theta}{n}}.
\end{aligned}
\tag{128}
$$

hence

$$\frac{\varphi_+(\theta; \xi, n)}{\varphi_-(\theta; \xi, n)} = \prod_{k=1}^{[\frac{1}{2\xi}]} \frac{\cos\left(\frac{\pi(1-2k\xi)}{n}\right) + 1}{\cos\left(\frac{\pi - 2\pi k\xi}{n}\right) - \cosh\frac{\theta}{n}}, \tag{129}$$

that is (127) holds.

Let us now turn to the issue of the dynamical pole axiom (33) and write down some identities involving the ratios of the minimal form factors $R(\theta; \xi, n)$ and $G_{\pm}(\theta; xi, n)$. Restricting ourselves first to the regime where the second breather is already present $\frac{1}{2} \geqslant \xi$, we can evaluate the residue in $F_{s\bar{s}}(\theta; \xi, n)$ corresponding to the second breather as

$$-i\underset{\theta=0}{\text{Res}} F_{s\bar{s}}(\theta + i\pi(1 - 2\xi); \xi, n) = -\langle \mathcal{T} \rangle \sin\frac{\pi}{n} \csc\frac{\pi(1 - 2\xi)}{n} \frac{G_-(i\pi(1 - 2\xi); \xi n)}{G_-(i\pi; \xi n)}. \tag{130}$$

This function is compared to

$$\Gamma^{b_2}_{s\bar{s}}F_{b_2}(\xi,n) = \frac{\sqrt{\sin(2\pi\xi)}\,\csc^2\frac{\pi\xi}{2}}{2} \frac{\sin\frac{\pi}{n}\sqrt{2\tan\pi\xi}\,R(-i\pi\xi;\xi,n)}{\left(2n\sinh\left(\frac{i\pi(1-\xi)}{2n}\right)\sinh\left(\frac{i\pi(\xi+1)}{2n}\right)\right)R(i\pi;\xi,n)}$$

$$= \csc^2\frac{\pi\xi}{2}\frac{\langle\mathcal{T}\rangle\sin\frac{\pi}{n}\sin\pi\xi\,R(-i\pi\xi;\xi,n)}{2n\sinh\frac{i\pi(1-\xi)}{2n}\sinh\frac{i\pi(\xi+1)}{2n}R(i\pi;\xi,n)}\ . \tag{131}$$

This means, that the following identity holds

$$\frac{n\tan\frac{\pi\xi}{2}\sin\frac{\pi(1-\xi)}{2n}\sin\frac{\pi(1+\xi)}{2n}}{\sin\frac{\pi(1-2\xi)}{n}} = \frac{R(-i\pi\xi;\xi,n)G_-(i\pi;\xi,n)}{R(i\pi;\xi,n)G_-(i\pi(1-2\xi);\xi,n)} \tag{132}$$

for any integer $n\geqslant 1$ and $\frac{1}{2}\geqslant\xi$, which can be easily verified numerically.

Concerning now the regime when the breather $b_4$ is present, that is, $\frac{1}{4}\geqslant\xi>0$, we write

$$-i\mathop{\mathrm{Res}}_{\theta=0}F_{s\bar{s}}(\theta+i\pi(1-4\xi);\xi,n) = \langle\mathcal{T}\rangle\frac{2\sin\frac{\pi}{n}\sin\left(\frac{\pi\xi}{n}\right)\sin\left(\frac{\pi(1-\xi)}{n}\right)\csc\left(\frac{\pi(1-4\xi)}{n}\right)}{\cos\left(\frac{\pi(1-4\xi)}{n}\right)-\cos\left(\frac{\pi(1-2\xi)}{n}\right)}\frac{G_-(i\pi(1-4\xi);\xi,n)}{G_-(i\pi;\xi,n)} \tag{133}$$

which is expected to be equal to

$$\Gamma^{b_4}_{s\bar{s}}F_{b_4}(\xi,n) = \langle\mathcal{T}\rangle\Gamma^{b_4}_{s\bar{s}}\Gamma^{b_3}_{b_3b_1}\Gamma^{b_3}_{b_2b_1}\Gamma^{b_2}_{b_1b_1}\frac{\sin\frac{\pi}{n}\sin\frac{\pi}{2n}(1+2\cos\frac{\pi\xi}{n})\cos\frac{\pi(1-\xi)}{2n}}{2n^2\sin^2\frac{\pi(1+\xi)}{2n}\sin\frac{\pi(1-2\xi)}{2n}\sin\frac{\pi(1-3\xi)}{2n}}$$

$$\times\frac{R(-3\pi i\xi;\xi,n)R(-2\pi i\xi;\xi,n)^2R(-i\pi\xi;\xi,n)^3}{R(i\pi;\xi,n)^2}$$

$$= \langle\mathcal{T}\rangle\frac{2\sin\frac{\pi}{2n}\sin\frac{\pi}{n}\cot^2\frac{\pi\xi}{2}\cos\frac{\pi(1-\xi)}{2n}\left(1+2\cos\frac{\pi\xi}{n}\right)}{n^2\sin^2\frac{\pi(1+\xi)}{2n}\sin\frac{\pi(1-2\xi)}{2n}\sin\frac{\pi(1-3\xi)}{2n}}$$

$$\times\frac{R(-3\pi i\xi;\xi,n)R(-2\pi i\xi;\xi,n)^2R(-i\pi\xi;\xi,n)^3}{R(i\pi;\xi,n)^2}\ . \tag{134}$$

This means, that the following identity holds

$$\frac{n^2\tan^2\frac{\pi\xi}{2}\sin\frac{\pi\xi}{n}\sin^2\frac{\pi(1+\xi)}{2n}\sin\frac{\pi(1-3\xi)}{2n}\sin\frac{\pi(1-2\xi)}{2n}\sin\frac{\pi(1-\xi)}{n}}{\sin\frac{\pi}{2n}\left(1+2\cos\frac{\pi\xi}{n}\right)\left(\cos\frac{\pi(1-4\xi)}{n}-\cos\frac{\pi(1-2\xi)}{n}\right)\sin\frac{\pi(1-4\xi)}{n}\cos\frac{\pi(1-\xi)}{2n}}$$

$$= \frac{R(-3\pi i\xi;\xi,n)R(-2\pi i\xi;\xi,n)^2R(-i\pi\xi;\xi,n)^3G_-(i\pi;\xi n)}{R(i\pi;\xi,n)^2G_-(i\pi(1-4\xi);\xi n)}\ , \tag{135}$$

for any integer $n\geqslant 1$ and $\frac{1}{4}\geqslant\xi$, which can be verified numerically. Indeed, these two identities clearly hold when checked numerically, an analytic proof, however, has not yet been achieved.

# E   $\Delta$ Sum Rule Evaluation

In this Appendix we summarize our numerical results for the sum (72) and several distinct values of $\xi$ and $n$. As discussed in Section 6 we include one- and two-particle contributions. In the regime $1 \geqslant \xi > \frac{1}{3}$ all the non-vanishing two- and one-particle contributions are taken into account as described in equations (73)-(74).

As we can see in Tables 1 and 2, the contribution from the $b_1 b_1$ and $b_2 b_2$ terms is very small compared to those of $s\bar{s}$ and $b_2$. Assuming this tendency to hold for the contributions $b_3 b_3$, $b_2 b_4$ and $b_4 b_4$, in the interaction regimes $\frac{1}{3} \geqslant \xi > \frac{1}{5}$ we have neglected the corresponding terms and still found good saturation of the rule (see Table 3).

| $n$ | $\Delta_{\mathcal{T}}$ | $s\bar{s}$ | $b_1 b_1$ | $b_2 b_2$ | $b_2$ | $\sum$ |
|---|---|---|---|---|---|---|
| 2 | 0.0625 | 0.0526252 | 0.0026597 | 0.0000016 | 0.0050643 | 0.0603508 |
| 3 | 0.11111 | 0.0930742 | 0.0049999 | 0.0000030 | 0.0085415 | 0.1066187 |
| 4 | 0.15625 | 0.1306165 | 0.0071536 | 0.0000044 | 0.0117942 | 0.1495687 |
| 5 | 0.2 | 0.1670215 | 0.0092264 | 0.0000057 | 0.0149699 | 0.1912235 |

(a) $\xi = 0.48734$

| $n$ | $\Delta_{\mathcal{T}}$ | $s\bar{s}$ | $b_1 b_1$ | $b_2 b_2$ | $b_2$ | $\sum$ |
|---|---|---|---|---|---|---|
| 2 | 0.0625 | 0.0398813 | 0.0034363 | 0.0000204 | 0.0168508 | 0.0601887 |
| 3 | 0.11111 | 0.0712682 | 0.0064312 | 0.0000387 | 0.0285433 | 0.1062814 |
| 4 | 0.15625 | 0.1003544 | 0.0091893 | 0.0000555 | 0.0394690 | 0.1490682 |
| 5 | 0.2 | 0.1285211 | 0.0118452 | 0.0000717 | 0.0501288 | 0.1905668 |

(b) $\xi = 0.45133$

| $n$ | $\Delta_{\mathcal{T}}$ | $s\bar{s}$ | $b_1 b_1$ | $b_2 b_2$ | $b_2$ | $\sum$ |
|---|---|---|---|---|---|---|
| 2 | 0.0625 | 0.0230446 | 0.0054990 | 0.0000904 | 0.0313208 | 0.0599548 |
| 3 | 0.11111 | 0.0419772 | 0.0102062 | 0.0001745 | 0.0534327 | 0.1057905 |
| 4 | 0.15625 | 0.0594778 | 0.0145467 | 0.0002518 | 0.0740622 | 0.1483385 |
| 5 | 0.2 | 0.0763850 | 0.0187308 | 0.0003260 | 0.0941674 | 0.1896092 |

(c) $\xi = 0.38231$

| $n$ | $\Delta_{\mathcal{T}}$ | $s\bar{s}$ | $b_1 b_1$ | $b_2 b_2$ | $b_2$ | $\sum$ |
|---|---|---|---|---|---|---|
| 2 | 0.0625 | 0.0112334 | 0.0093408 | 0.0001931 | 0.0390784 | 0.0598457 |
| 3 | 0.11111 | 0.0209003 | 0.0171719 | 0.0003779 | 0.0671062 | 0.1055563 |
| 4 | 0.15625 | 0.0298140 | 0.0244039 | 0.0005479 | 0.0932251 | 0.1479910 |
| 5 | 0.2 | 0.0384043 | 0.0313831 | 0.0007111 | 0.1186557 | 0.1891542 |

(d) $\xi = 0.30091$

Table 2:   One- and two-particle contributions to the $\Delta$ sum rule for four values of $\xi \in (\frac{1}{3}, \frac{1}{2})$. It is interesting to observe how the breather contributions become larger as $\xi$ is decreased, sending the theory deeper into the attractive regime. For instance, in Table (d) the $s\bar{s}$ contribution accounts only for 20% of the value of $\Delta_{\mathcal{T}}$.

| $n$ | $\Delta_\mathcal{T}$ | $s\bar{s}$ | $b_1 b_1$ | $b_2 b_2$ | $b_1 b_3$ | $b_2$ | $b_4$ | $\sum$ |
|---|---|---|---|---|---|---|---|---|
| 2 | 0.0625 | 0.0031453 | 0.0154695 | 0.0002683 | 0.0004363 | 0.0395954 | 0.0015294 | 0.060444 |
| 3 | 0.11111 | 0.0060694 | 0.0281816 | 0.0005309 | 0.0008206 | 0.0683090 | 0.0027951 | 0.106707 |
| 4 | 0.15625 | 0.0087587 | 0.0399377 | 0.0007727 | 0.0011717 | 0.0950521 | 0.0039606 | 0.149653 |
| 5 | 0.2 | 0.0113407 | 0.0512960 | 0.0010044 | 0.0015091 | 0.1210735 | 0.0050857 | 0.191309 |

Table 3: One- and two-particle contributions to the $\Delta$ sum rule for $\xi = 0.22108$. For this value of the coupling the first four breathers can be formed and approximately half the value of $\Delta_\mathcal{T}$ comes from breather contributions. Even after neglecting the terms $b_3 b_3, b_2 b_4$ and $b_4 b_4$ the rule is 95% saturated.

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
