# Peer review of "Branch Point Twist Field Form Factors in the sine-Gordon Model I: Breather Fusion and Entanglement Dynamics"

_SciPost Physics_

## Round 1 · Referee Report · Anonymous (Referee 1) · 2021-5-4

Strengths

see report

Weaknesses

see report

Report

The paper "Branch Point Twist Field Form Factors in the sine-Gordon Model I:
Breather Fusion and Entanglement Dynamics" by O. A. Castro-Alvaredo and D.X. Horvath determines the form factors built up from few Breather excitations for the branch point twist field operator in the replica-cpopied Sine-Gordon model. The results obtained by the authors are tested by means of the so-called $\Delta$ sum rule. The obtained form factors allows the authors to carry out a qualitative analysis of the time dependence of the Renyi entropy after a small mass quench in the system.

In overall, the paper is well written and easy to read. The scope of techniques used in the paper is also up to date in respect to the ongoing research relative to correlation functions in interacting integrable models. I find it a pity that the authors did not compute all the form factors of this field, for instance by using the representations of Babujian et al., but rather limited themselves to
only the ones involving the first few Breather excitations. Doing the general case should, effectively speaking, only amount to identifying the appropriate p functions for their fields of interest. Despite this criticism, the technical results obtained authors already allowed them for some applications to a physically interesting situation. Therefore, provided that the below points are met, I recommend the paper for publication.

Requested changes

(i) programme $\rightarrow$ program

(ii) It would be reasonable if the authors precised that that only compute low-particle number form factors (in the title/abstract/conclusion, for instance).

(iii) To a certain extent, I disagree with the statement made in the introductions which distinguishes between the approached of Smirnov and Babujian et al. Intrinsically speaking, the procedures are the same but simply use different bases of the Hilbert space to decompose the form factors. In both cases, these are given by off-shell Bethe vectors, evaluated at the inhomogeneities in Smirnov's case and evaluated at generic parameters in Babujian et al's case. See also [11] where it is suggested to use on-shell Bethe vectors for the decomposition of form factors.

(iv) While reference [10] in the introduction is definitely a reference for the Breather form factors, the procedure was known in the 80s, see the extensive work of Smirnov on Sine-Gordon.

(v) To the references [22, 23] one could add those to the works of Karowski et al (H.~Babujian and M.~Karowski, Sine-Gordon breather form factors and quantum field equations, J.Phys.A, 35, (2002), 9081-9104.) and Lukyanov et al.(V.~Brazhnikov and S.~Lukyanov,Angular quantization and form factors in massive integrable models , Nucl. Phys. B, 512, 3, (1998), 616-636).

(vi) (135) It should be possible to prove this formula by, first, expressing R in terms of a ratio of four quantum dilogarithms. Likewise, the ratio of $G_-$ functions will reduce to some products of ratios of quantum dilogarithms. Then, it should only remain to use the finite difference equation satisfied by the quantum dilogarithm so as to cancel eventually out all the dilogarithms and get the ratios of trigonometric functions.

  • validity: ok
  • significance: good
  • originality: ok
  • clarity: high
  • formatting: excellent
  • grammar: excellent

Author:  David Horvath  on 2021-05-22  [id 1457]

(in reply to Report 1 on 2021-05-04)

We thank the referee for their constructive suggestions. Below we answer each point separately and explain what changes we have made to the paper.

(i) programme → program

Done!

(ii) It would be reasonable if the authors precised that that only compute low-particle number form factors (in the title/abstract/conclusion, for instance).

We have now included the wording “low particle-number” in the abstract and at the beginning of the conclusion. In the introduction, our computations are only really discussed in the last two paragraphs and in there, there was already a rather precise description of which form factors we compute, so we have kept the introduction unchanged in this respect. We think the title is best kept in its current form as it is already quite long. In addition, the paper discusses some features of the form factors (i.e. their general properties and equations) which are more general than just low particle numbers and the title reflects this too.

(iii) To a certain extent, I disagree with the statement made in the introductions which distinguishes between the approached of Smirnov and Babujian et al. Intrinsically speaking, the procedures are the same but simply use different bases of the Hilbert space to decompose the form factors. In both cases, these are given by off-shell Bethe vectors, evaluated at the inhomogeneities in Smirnov's case and evaluated at generic parameters in Babujian et al's case. See also [11] where it is suggested to use on-shell Bethe vectors for the decomposition of form factors.

We have reworded this paragraph slightly employing the referee’s suggestions.

(iv) While reference [10] in the introduction is definitely a reference for the Breather form factors, the procedure was known in the 80s, see the extensive work of Smirnov on Sine-Gordon.

We included a comment to highlight this.

(v) To the references [22, 23] one could add those to the works of Karowski et al (H.~Babujian and M.~Karowski, Sine-Gordon breather form factors and quantum field equations, J.Phys.A, 35, (2002), 9081-9104.) and Lukyanov et al.(V.~Brazhnikov and S.~Lukyanov,Angular quantization and form factors in massive integrable models , Nucl. Phys. B, 512, 3, (1998), 616-636).

Done.

(vi) (135) It should be possible to prove this formula by, first, expressing R in terms of a ratio of four quantum dilogarithms. Likewise, the ratio of G−G− functions will reduce to some products of ratios of quantum dilogarithms. Then, it should only remain to use the finite difference equation satisfied by the quantum dilogarithm so as to cancel eventually out all the dilogarithms and get the ratios of trigonometric functions.

We thank the referee for this helpful suggestion. We are not very familiar with the use of quantum dilogarithms in this context, but we have reconsidered the identities (132) and (135) (in the original version) and have now succeeded in proving both. We have found a way to prove them using mainly standard properties of Gamma-functions and the Gamma-function representation of minimal form factors. The new proofs have been added in Appendix D.

---

## Round 1 · Referee Report · Anonymous (Referee 2) · 2021-5-18

Report

The paper is devoted to the calculation of form factors of the branch point twist field in the sine-Gordon model. The relevance of the form factors of this field comes from the fact that they can allow the evaluation of the entanglement entropy along the lines shown by Cardy, one of the authors and Doyon in Ref. [24]. While the latter work had already considered the repulsive regime of the sine-Gordon model, the present paper extends the calculations to the attractive regime in which solitons and antisolitons produce bound states (breathers). The authors confirm the consistency of their results for the twist field form factors using the cluster property at large rapidity separations and the $\Delta$ sum rule for the conformal dimension. This analysis extends to the case of the twist field that performed for the ordinary sine-Gordon fields in Ref. [G. Delfino, P. Grinza, NPB 682 (2004) 521], a paper that needs to be quoted.

As a particularly interesting application of their results for the twist field form factors the authors determine the time evolution of the entanglement entropy following a quantum quench of the sine-Gordon mass in the attractive regime. The calculation is performed using the theory of quantum quenches derived by Delfino in Ref. [45], which predicts undamped oscillations of one-point functions when the one-particle form factor is not zero. The authors show that the prediction holds also for the entanglement entropy of a semi-infinite region, which is related to the one-point function of the twist field. Being in the attractive regime is essential since the undamped oscillations are due to breathers.

I add two other points for revision to the one I already mentioned:

  1. In the Introduction, the review of previous results for sine-Gordon form factors should include the paper [G. Delfino, Phys. Lett. B 450 (1999) 196], where matrix elements of the order/disorder fields were first obtained.

  2. There is a typo on page 19 just above the paragraph “We demonstrate …”: the regime of damped oscillations is $\xi>1/2$ (not $\xi<1/2$).

  • validity: -
  • significance: -
  • originality: -
  • clarity: -
  • formatting: -
  • grammar: -

Author:  David Horvath  on 2021-05-22  [id 1458]

(in reply to Report 2 on 2021-05-18)

We thank the referee for their positive assessment of the paper and their suggestions.

The paper is devoted to the calculation of form factors of the branch point twist field in the sine-Gordon model. The relevance of the form factors of this field comes from the fact that they can allow the evaluation of the entanglement entropy along the lines shown by Cardy, one of the authors and Doyon in Ref. [24]. While the latter work had already considered the repulsive regime of the sine-Gordon model, the present paper extends the calculations to the attractive regime in which solitons and antisolitons produce bound states (breathers). The authors confirm the consistency of their results for the twist field form factors using the cluster property at large rapidity separations and the Δ sum rule for the conformal dimension. This analysis extends to the case of the twist field that performed for the ordinary sine-Gordon fields in Ref. [G. Delfino, P. Grinza, NPB 682 (2004) 521], a paper that needs to be quoted.
We have now cited the paper at the end of subsection 5.3 and also in the introduction.
As a particularly interesting application of their results for the twist field form factors the authors determine the time evolution of the entanglement entropy following a quantum quench of the sine-Gordon mass in the attractive regime. The calculation is performed using the theory of quantum quenches derived by Delfino in Ref. [45], which predicts undamped oscillations of one-point functions when the one-particle form factor is not zero. The authors show that the prediction holds also for the entanglement entropy of a semi-infinite region, which is related to the one-point function of the twist field. Being in the attractive regime is essential since the undamped oscillations are due to breathers.
I add two other points for revision to the one I already mentioned:
• In the Introduction, the review of previous results for sine-Gordon form factors should include the paper [G. Delfino, Phys. Lett. B 450 (1999) 196], where matrix elements of the order/disorder
We have cited this paper and the one above in the second paragraph of the introduction.
• There is a typo on page 19 just above the paragraph “We demonstrate …”: the regime of damped oscillations is ξ>1/2 (not ξ<1/2).
Noted and corrected.

---

## Editorial Decision

resubmitted